# MUEDIT: A LIGHTWEIGHT YET EFFECTIVE *Mu*LTI-TASK MODEL *Editing* METHOD

## ABSTRACT

Existing model editing methods encounter limitations in handling multi-task knowledge updates, primarily due to interference between different tasks. To address this gap, this paper first provides a formal definition of multi-task editing, which is different from traditional sequential editing, and subsequently analyzes the shortcomings of traditional editing methods and Fang's null-space projection method, which fails to generalize to multi-task scenarios. To tackle this challenge, we introduce a novel concept termed the **Conflict Index**, which quantifies the degree of conflicts between the editing objectives of two tasks. Building on this index, We then design two strategies to mitigate multi-task conflicts: (1) **identifying the optimal editing path** that minimizes the total conflict index across all tasks, and **adopting a low-rank matrix approximation method based on the conflict index to expand the null-space dimension** when conflicts remain high. Experimental results show that our proposed Mu-Edit method effectively alleviates multi-task editing conflicts. It outperforms existing baseline methods across various evaluation metrics on multiple tasks while preserving the model's capabilities in general domains.

## 1 INTRODUCTION

Large language models (LLMs) have recently demonstrated outstanding performance in diverse areas such as natural language understanding (Dušek et al., 2020), mathematical reasoning (Imani et al., 2023), and knowledge-intensive question answering (Sun et al., 2024). However, despite their impressive capabilities, LLMs remain prone to misinterpreting human instructions and generating incorrect or outdated responses (Bai et al., 2024; Chen et al., 2024). This has spurred exploration into model editing and various continuous learning techniques aimed at refining LLMs' behavior over time (Ji et al., 2024; Wang & Li, 2024).

In addition to directly fine-tuning LLMs on specific tasks, recent studies have introduced model editing techniques to enable LLMs to discard specific erroneous knowledge while preserving their overall functionality. Building upon this concept, several model editing methods have emerged. ROME (Meng et al., 2022a) uses a logit attribution method to identify the location of knowledge and then edits it by updating specific factual associations. MEMIT (Meng et al., 2022b) is another effective method that locates knowledge and directly updates

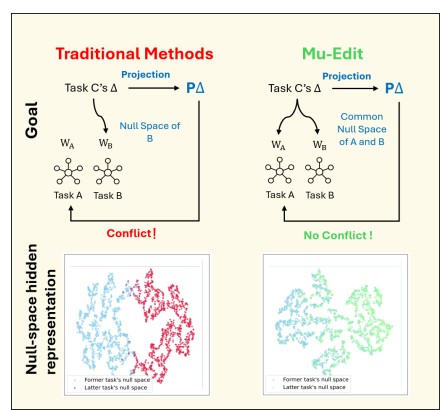

Figure 1: Comparison of Goal and Null-Space Hidden Representation with our Mu-Edit and existing methods. Existing method's have almost no common null space between tasks, leading to severe conflicts, while our Mu-Edit expands the common null space obviously.

large-scale memories. Some methods edit models without explicit localization, such as the approach proposed by Ni et al. (2023), which introduces a "forgetting before learning" paradigm: LLMs are first trained to forget incorrect answers before learning new information, leading to improved per-

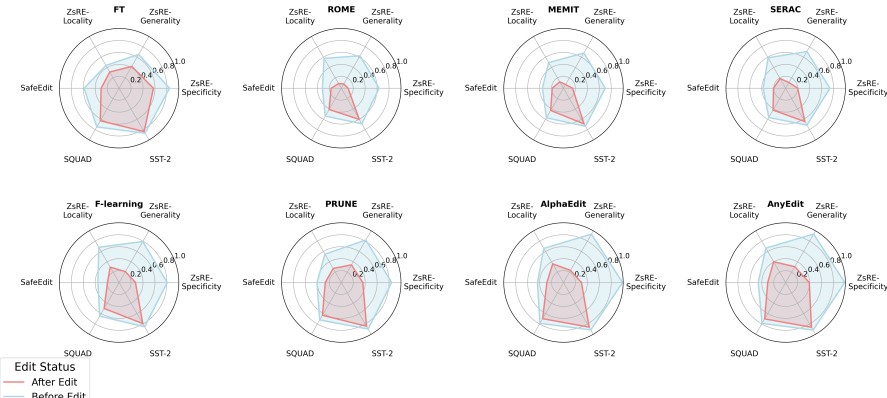

Figure 2: Illustration of existing models' multi-task editing performance decline; For existing model editing methods, we choose five tasks, namely ZsRE, SafeEdit, SQUAD, SST-2 and CKnowEdit and compare all method's performance on the target task before and after editing on other tasks. All metrics are higher the better.

formance compared to direct fine-tuning. However, current model-editing methods face limitations in maintaining performance across multiple edits and generalizing to multi-task knowledge updates simultaneously. Although some works, such as those by Ma et al. (2024); Fang et al. (2024), have alleviated interference from multiple edits within a single task by restricting the number of conditions for matrix updates or projecting to a null space, they fail to address the multi-task editing interference problem (Li et al., 2025a;c).

In contrast to these existing studies, the paper introduces a novel paradigm: **Multi-task model editing**. We first provide a formal the definition of multi-task editing, and then clarify its core goal: To update knowledge in different tasks simultaneously without interfering the performance of other tasks. This formulation is motivated by the critical challenge that existing editing methods suffer from severe performance degradation when updating multi-task knowledge simultaneously as shown in Fig 2, which illustrates that current editing methods have caused serious model collapse for multi-task editing. They often forget the knowledge edited in the previous tasks during the editing process, resulting in substantial performance degradation, which become more pronounced as the number of tasks accumulates. Even though direct fine-tuning method mitigates model collapse to some extent compared to existing methods, it still exhibits a notable decline in performance. These observations collectively indicate that significant conflicts exist between the editing objectives of different tasks, and existing model editing methods struggle to effectively decouple these distinct editing objectives, hindering the ability to update knowledge across multiple tasks simutaneously.

To address this core issue, we firstly conducted an in-depth analysis based on Fang's null-space projection method, revealing that its failure to generalize to multi-task scenarios stems from a critical limitation: when updating knowledge across different tasks, **the null-space projection matrix of the current task may not necessarily lie within the null space of previously edited tasks**. Building on this insight, we propose a novel approach: If we project the editing parameters onto the common null space shared by the current task and the preceding editing task during multi-task model editing, we can theoretically reduce conflicts arising from competing task-specific editing objectives. To operationalize this idea, We propose a new concept based on this goal——**Conflict Index**. Leveraging this index, we developed a novel framework called **Mu-Edit**, which incorporates two complementary strategies to resolve multi-task conflicts: 1. Determining the optimal editing sequence by minimizing the total conflict index across all tasks. 2. Designing a conflict-index-guided low-rank matrix approximation method to actively expand the null-space dimension. And the selection as of which task's $K$ to approximate is determined by the sum of the total conflict index with all other tasks. Experiments verify that our Mu-Edit can greatly alleviate the conflict issues caused by multi-task editing in already known task numbers and types. For increasing task numbers and types, we also develop a method called **Greedy Search** to handle the issue of excessive computational complexity of best order search and editing on unpredictable task types.

Generally speaking, our main contributions are three-folds:

1. To the best of our knowledge, this work is the first to investigate multi-task knowledge editing **from the perspective of null-space conflicts**. We further propose the concept of **Conflict Index** based on the null-space properties of task-corresponding matrices and enables effective quantification of the conflict degree between different tasks.

2. Building on this Conflict Index, We develop two complementary optimization strategies: (1) Computing the optimal editing sequence and performing edits in accordance with this sequence; (2) Expanding the null space dimension and reducing multi-task conflicts by applying low-rank approximation on the original matrix.

3. Experimental results confirm that, compared with existing model editing methods, our proposed approach not only achieves a significant improvement in multi-task editing performance, but also effectively preserves the model's capabilities in general domains.

## 2 PRELIMINARY

### 2.1 KNOWLEDGE STORAGE AND EDITING IN LLMS

Assuming that the hidden state of the $i$-th layer for a specific token as $h^i \in \mathbb{R}^d$, the multi-layer perceptron (MLP) module within the $i$-th layer can then be described as follows:

$$h^i = \sigma(\tilde{h}^i W_1^i) \cdot W_2^i,$$

where $W_1^i$ and $W_2^i$ represent trainable parameters of transition matrix, $\tilde{h}^i$ represents output of $i$-th MHA layer and $\sigma(\cdot)$ denotes the activation function. Following prior works (Meng et al., 2022a;b), we express the attention block and MLP in parallel. The MLP layers can be interpreted as linear associative memory (Geva et al., 2021).

### 2.2 TRADITIONAL MODEL-EDITING METHODS

Model editing aims to update knowledge stored in LLMs through a single edit or multiple edits (*i.e.*, sequential editing). In each edit within the locate-then-edit paradigm, we modify the model parameters $W$ by adding a perturbation $\Delta$. Specifically, the knowledge stored in the model can be formalized as triplets $(s, r, o)$, each edit needs to update $u$ pieces of knowledge in the form of $(s, r, o)$, where $s$, $r$ and $o$ means subject, relation and object seperately. The new parameter $W$ is expected to associate new $k$-$v$ pairs, where $k$ encodes knowledge component of $(s, r)$ and $v$ encodes $(o)$. $W$ is in the dimension of $d_o * d_i$, where $d_i$ and $d_o$ represent the dimensions of the FFN's intermediate and output layers. We can define the knowledge matrix of all $k$-$v$ pairs as follows:

$$K_0 = [k_1, k_2, \dots, k_u] \in \mathbb{R}^{d_i \times u}, V_0 = [v_1, v_2, \dots, v_u] \in \mathbb{R}^{d_o \times u}$$

Where $u$ represents the scale of knowledge pairs, and the subscripts of $k$ and $v$ represent the index of the to-be-updated knowledge.

ROME (Meng et al., 2022a) proposes that the objective of model editing is optimized by the following equation, which edits model by minimizing the distances of selected key-vectors before and after editing, while memorizing a new $k$-$v$ pair, which is $k_e$-$v_e$:

$$\hat{W} = \arg \min_{\hat{W}\,:\,\hat{W} k_e = v_e} \underbrace{\|\hat{W} K_0 - W_0 K_0\|_F^2}_{\text{Preservation}}$$

Although ROME is effective for single sample editing, it cannot be used to edit multiple samples simultaneously, and the later method MEMIT (Meng et al., 2022b) solves this problem. MEMIT on the other hand optimizes a relaxed version of the same objective:

$$\hat{W} = \operatorname*{argmin}_{\hat{W}} \; \lambda \underbrace{\left\|\hat{W} K_0 - W_0 K_0\right\|_F^2}_{\text{preservation}} + \underbrace{\left\|\hat{W} K_E - V_E\right\|_F^2}_{\text{memorization}}$$

where $K_E = [k_1^e \,|\, k_2^e \,|\, \dots \,|\, k_E^e]$ is a matrix containing a row of vectors representing the edits we make in a batch and $V_E = [v_1^e \,|\, v_2^e \,|\, \dots \,|\, v_E^e]$ represents their target representations. The above optimization objective aims to modify the output representations of vectors in $K_E$ to $V_E$ by minimizing the least square error between them instead of requiring them to be equal through an equality constraint.

## 2.3 Sequential-optimized editing methods

PRUNE (Ma et al., 2024) and AlphaEdit (Fang et al., 2024) have provided different solutions to the problem of model collapse caused by traditional multiple editing methods.

PRUNE designs experiments and finds that model collapse of multiple editing mainly comes from the gradually increasing maximum singular value of $\Delta W$. On the other hand, AlphaEdit addresses multiple editing model collapse through null-space projection. Null-space is defined as follows: given two matrices $A$ and $B$, $B$ is in the null space of $A$ if and only if $BA = 0$. Fang defines $\bar{\mathbf{K}}_{t-1} = [\mathbf{K}_1; \cdots; \mathbf{K}_{t-1}]$ represent the keys of all previous editing steps, and $\bar{\mathbf{V}}_{t-1} = [\mathbf{V}_1; \cdots; \mathbf{V}_{t-1}]$ represent values from all previous update steps. Fang proposes a new method to mitigate the negative interference. He restricts the updates constrained to lie within the null space of the previously injected knowledge representations. To say it specifically, Fang proposes that the perturbation matrix $\Delta$ should be projected onto the null space of $\bar{\mathbf{K}}_{t-1}$ so we can obtain an equation:

$$(\mathbf{\Delta W}_t + \mathbf{W}_{t-1})\bar{\mathbf{K}}_{t-1} = \mathbf{W}_{t-1}\bar{\mathbf{K}}_{t-1} = \bar{\mathbf{V}}_{t-1}$$

This implies that the projection $\Delta$ will not disrupt the key-value associations of previous updated knowledge and ensure we only focus on the new knowledge to be updated.

# 3 Methods

## 3.1 Definition of multi-task editing

In multi-task editing, supposing we aim to update knowledge across $N$ distinct tasks, and input data from each task is described as $I_1, I_2, ..., I_N$. Initial parameters before training on $\mathbf{I}_n$ ($n \in \{1, ..., N\}$) are initialized as $\mathbf{W}_{n-1}$, which are the optimal parameters obtained after training on the previous data $\mathbf{I}_{n-1}$. And after editing on the task $n$, we define the model parameters as $\mathbf{W}_n$. Once all $N$ tasks are edited, the final model parameters are defined as $\mathbf{W}^*$. A core premise of multi-task editing is task independence, a property that fundamentally distinguishes it from scenarios involving "single-task sub-datasets" (e.g., ZsRE and Counterfact).

## 3.2 Rethinking of the null space of multiple tasks

We have conducted deeper thinking based on Fang et al. (Fang et al., 2024)'s definition: theoretically, if the editing knowledge of the $n$-th task can be projected onto the common null space formed by all previous edited tasks, AlphaEdit's method should be directly adaptable to multi-task editing scenarios. However, our experimental results demonstrate that such direct adaptation fails to guarantee satisfactory performance. We attribute the limitation to a key observation: During sequential multi-task editing, the new knowledge matrix $K_n$ compresses the null space of $K_{n-1}$. To elaborate, in an ideal conflict-free scenario, the updated model parameter matrix $W_{n+1}$ should be projected onto the common null space of $K_n$ and $K_{n-1}$. Yet, the common null space of the vertically concatenated matrix of $[K_n^T; K_{n-1}^T]$ is no larger than the smallest null space in $K_n$ and $K_{n-1}$ (we will give a proof). As model editing proceeds across more tasks, this null-space compression becomes increasingly severe——ultimately leading to a noticeable decline in editing performance.

To address the aforementioned issue, we first observe that the knowledge matrix $K$ of different tasks induces varying degrees of null-space compression during the editing process. We hypothesize that editing tasks in a sequence that minimizes the null space compression could yield improved performance. To operationalize this hypothesis, we first define a null-space conflict metric to quantify the conflict between tasks $i$ and $j$, which is called **Conflict Index**. Specifically, $\mathbb{N}(K_{il})$ denotes the null space of knowledge matrix of task $i$ at layer $l$, and $\mathbb{N}\left(\left[K_{il}^T : K_{jl}^T\right]^T\right)$ denotes the null space of vertically-concatenated knowledge matrix of task $i$ and $j$ at layer $l$:

$$C(K_{il}, K_{jl}) = 1 - \frac{\dim\left(\mathbb{N}\left(\left[K_{il}^T : K_{jl}^T\right]\right)^T\right)}{\min\left\{\dim(\mathbb{N}(K_{il})), \dim(\mathbb{N}(K_{jl}))\right\}} \tag{1}$$

We will prove that the null space of column-combined matrix is equivalent to the common null space of $K_{il}$ and $K_{jl}$, but the latter is difficult to calculate, so we will use the column-combined calculation method. Since the conflict index of each layer may not be the same, we further average the conflict index of each layer to obtain the zero-space conflict index of the two tasks $i$ and $j$, so the conflict index can be calculated as follows:

$$C(K_i, K_j) = \frac{1}{L} \sum_{l \in [1,L]} C(K_{il}, K_{jl}) \tag{2}$$

We will demonstrate in the appendix that the conflict index can accurately reflect the degree of interference between parameter updates for two tasks, a larger conflict index indicates greater interference between the updates of the two tasks.

### 3.3 RESOLVING THE INTERFERENCE

#### 3.3.1 FINDING THE BEST EDITING ORDER

Building on the preceding definition of conflict index, we first quantify the pairwise interference between all task pairs, then determine the editing sequence that minimizes the total interference across all tasks. Formally, We define the optimal editing order as follows:

$$\textbf{Best Order}(K_N) = \min_{\sigma \in \mathcal{S}_N} \sum_{n=1}^{N-1} C(K_{\sigma(n)}, K_{\sigma(n+1)}). \tag{3}$$

Where $\mathcal{S}_N$ denotes the symmetric group containing all permutations of $N$ tasks. Each $\sigma \in \mathcal{S}_N$ represents a complete ordering of the tasks, and the optimal sequence is selected by minimizing the total cost among all possible permutations.

#### 3.3.2 INCREASING THE COMMON NULL SPACE THROUGH LOW-RANK MATRIX DECOMPOSITION

We hypothesize that the dimension of the common null space depends not only on the conflict index between the two tasks, but also on the rank of the each task's corresponding knowledge matrix $K$. Consequently, effectively reducing the rank of the original knowledge matrix $K$ can expand the common null space dimension——thereby mitigating inter-task conflicts. To achieve this rank reduction, we propose performing Singular Value Decomposition(SVD) on $K$, then approximating the original matrix using only the top few singular values and their corresponding vectors. Specifically:

$$K = \sum_{i=1}^{R} \sigma_i u_i v_i^{\mathrm{T}} \tag{4}$$

Where $R$ denotes the rank of matrix $K$. To determine the number of singular values to retain, we analyze two scenarios from the perspective of null-space conflict: 1. If $C(K_i, K_j)$ is no greater than a predefined threshold $\mu$, the definition of the conflict index implies that the ratio of $\frac{\dim(\mathbb{N}([K_{il} : K_{jl}]))}{\min\{\dim(\mathbb{N}(K_{il})), \dim(\mathbb{N}(K_{jl}))\}}$ is higher than $1 - \mu$. For instance, when $\mu$ is 0.2, the ratio is greater than 0.8, a value sufficient to satisfy the editing requirements of both tasks, so no additional rank reduction is needed. 2. If $C(K_i, K_j) > \mu$, the common null space of $K_i$ and $K_j$ is too limited to accommodate the editing needs of both tasks. In this case, we propose expanding the dimension of $\mathbb{N}(K)$ by reducing the rank of $K$. Specifically, we define the "conflict excess" as $C(K_i, K_j) - \mu$, a larger conflict excess indicates a greater need for rank reduction. For simplicity, we adopt a linear decay strategy, when the rank reduction magnitude is determined by $\alpha(C(K_i, K_j) - \mu)$ (with $\alpha$ as a tuning parameter). The dimensionality of the preserved singular values is thus calculated as follows:

$$d_\sigma = \begin{cases} R, & \text{if } C(K_i, K_j) \leq \mu \\ R - \alpha R(C(K_i, K_j) - \mu), & \text{if } C(K_i, K_j) > \mu \end{cases} \tag{5}$$

We also draw two variations of our main method: 1) **Mu-Edit$^-$**: This variant uses fixed values for hyperparameters $\mu$ and $\alpha$. In our primary experiments, we set $\mu = 0.2$ and $\alpha = 1$. 2) **Mu-Edit**: We design a dynamic threshold adjustment strategy for $\mu$: instead of using a fixed value, $\mu$ is determined based on the sum of conflict indexes between the current task and all previously considered tasks. The formula of selecting $\mu$ can be defined as follows:

$$\overline{C_i} = \frac{1}{N} \sum_{j=1}^{N} C(K_i, K_j) \quad \mu_{ij} = \min(\overline{C_i} + t \cdot \sigma_{C_i}, \overline{C_j} + t \cdot \sigma_{C_j}) \tag{6}$$

Where $t$ is a dynamic hyparameter that controls the extent to which variance influences the threshold (We set it 0.9 by default). Specifically, we first compute the mean and variance of the conflict index

between task $n$ and all other tasks. This strategy enables $\mu$ to adapt not only to the overall level of conflict level of the datasets but also to the variability in conflict levels among different tasks. Using the updated threshold, we then calculate $\overline{K}$ as an low-rank approximate representation of the original matrix $K$:

$$\overline{K} = \sum_{i=1}^{d_\sigma} \sigma_i u_i v_i^{\mathrm{T}} \tag{7}$$

We add the additional ablation studies about the role of mean and variance as well as hyperparameter selection in Tab 4 and Tab 11 in the Appendix. Through low-rank decomposition, subsequent experiments will verify that this approach effectively reduces the conflict index while improving performance. A remaining key question is to determine whether the matrix $K_i$ or the matrix $K_j$ should undergo low-rank approximation. Given the inherent characteristics of multi-task editing, conflict mitigation must address not only pairwise conflicts between two tasks but also conflicts between a given task and all other tasks awaiting editing. To this end, we determine which matrix to approximate via low-rank decomposition as follows:

$$\overline{K} = \begin{cases} K_i, & \text{if } \sum_{t \neq j} C(K_i, K_t) > \sum_{t \neq i} C(K_j, K_t) \\ K_j, & \text{otherwise} \end{cases} \tag{8}$$

We further introduce two ablation variants: **Mu-Edit(left)** and **Mu-Edit(right)**. Unlike the full Mu-Edit variant, Mu-Edit(left) always reduces the rank of $K_i$, while Mu-Edit(right) exclusively reduces the rank of $K_j$. After calculating the new low-rank decomposition matrix $\overline{K}$, we iteratively recalculate all conflict indices among this matrix and other matrices until all conflict indices fall below the threshold. Following the projection strategy in (Fang et al., 2024), we compute the projection matrix $P$, which projects onto the common null space of matrices $K_{n-1}$ and $K_n$, and update the original parameter $\Delta$ to $\Delta P$. In this work, we focus on minimizing the edit distance between consecutive three tasks. The objective function can be defined as:

$$\Delta = \arg \min_{\tilde{\Delta}} \Big( \left\| (W + \tilde{\Delta}P)K_{n+1} - V_{n+1} \right\|^2 + \left\| \tilde{\Delta}P \right\|^2 + \left\| \tilde{\Delta}PK_{n-1} \right\|^2 + \left\| \tilde{\Delta}PK_n \right\|^2 \Big) \tag{9}$$

In the appendix **A.13**. we further extend our analysis to minimizing the edit distance among four or more tasks and present the corresponding results. To calculate the parameter update $\Delta$, we first define the residual term $V_{n+1} - WK_{n+1}$ as $E$, then $\Delta$ can be represented as:

$$\Delta^* = EK_{n+1}^T P \Big( K_{n+1}K_{n+1}^T P + P + K_{n-1}K_{n-1}^T P + K_n K_n^T P \Big)^{-1} \tag{10}$$

We will prove the reversibility of $\Delta^*$ in the appendix section **A.2**.

### 3.4 Important Experimental details

**Datasets**: For datasets selection, we follow the datasets used in EasyEdit2 (Xu et al., 2025), and we choose five representative tasks categories, with one or two datasets per category. Specifically: For English common sense knowledge editing, we choose Counterfact (Meng et al., 2022a) and ZsRE (Wang et al., 2023), for detoxifying knowledge editing, we choose SafeEdit (Wang et al., 2024a), for reasoning-based knowledge editing, we choose SQuAD (Rajpurkar et al., 2016) and GSM8k (Cobbe et al., 2021), for sentiment analysis knowledge editing, we choose SST-2 (Socher et al., 2013) and for Chinese Phonetic knowledge editing, we choose CKnowEdit (Xu et al., 2025). By default we use 500 pieces of data for editing each task of knowledge. Given that the editing order of datasets may impact the final model editing performance, we set the default editing order across all experiments as ZsRE, SafeEdit, SQUAD, SST-2, and CKnowEdit.

**Evaluation Metrics**: In line with prior works Meng et al. (2022a); Fang et al. (2024), for datasets Counterfact, ZsRE and CknowEdit, we employ Specificity (efficiency success), Generalization (paraphrase success), Locality (neighborhood success) as evaluation metrics. For SafeEdit, we employ Harmful Rate, and For SQUAD, GSM8k and SST-2, we just use ACC (Accuracy) to measure the predictive correctness. Details of these metrics are described in the appendix in **A.3**.

**Other experimental details**: For fair comparison of previous works, we employ seven baseline methods suitable for multi-edit scenarios: 1. ROME (Meng et al., 2022a). 2. MEMIT (Meng et al., 2022b), 3. SERAC (Mitchell et al., 2021) 4. F-learning (Ni et al., 2023) 5. PRUNE (Ma et al., 2024) 6. AlphaEdit (Fang et al., 2024). 7. AnyEdit (Jiang et al., 2025). We also compare the results with direct fine-tuning (FT). All the methods are evaluated three two backbone models: Llama3-8B

, Qwen2-7B and GPT2-xl, with edits performed on specific layers for each model: Llama3-8B and Qwen2-7B: layers [4, 5, 6, 7, 8], GPT2-xl: layers [13, 14, 15, 16, 17]. For constructing the reference knowledge matrix $K$ for each domain, we sample 100000 pieces of data from the training sets of the five tasks. If the scale of the original dataset fails to reach 100,000, we need to perform data augmentation by randomly replacing the elements in the (s, r, o) triplets. The results for GPT2-xl and Qwen2-7B are displayed in the Appendix in Tab 4 and Tab 8.

# 4 MAIN EXPERIMENTS AND RESULTS

## 4.1 MAIN RESULTS AND ABLATION STUDY RESULTS

Table 1: Comparison results of immediate tests and final tests of our methods and baseline methods under default editing order and out calculated best editing order. In this test we use Llama3-8B as backbone model. And I- means Immidiate test, F- means Final test, DO means default editing order, BO means best editing order. * means the improvement passes significance via t-test with $p < 0.05$ in five-times repetition compared to AnyEdit.

| | ZsRE | | | | | | | | |
|---|---|---|---|---|---|---|---|---|---|
| Method | I-Specificity↑ | I-Generality↑ | I-Locality↑ | F-Specificity↑ | F-Generality↑ | F-Locality↑ | Spec change↑ | Gen change↑ | Loc change↑ |
| AlphaEdit(DO) | 0.9891 | 0.9352 | 0.6606 | 0.3009 | 0.2351 | 0.3608 | -0.6882 | -0.6343 | -0.2998 |
| AnyEdit(DO) | 0.9899 | 0.9382 | 0.6650 | 0.3954 | 0.3061 | 0.4021 | -0.5945 | -0.6321 | -0.2629 |
| Mu-Edit(DO) | 0.9883 | 0.9366 | 0.6669 | 0.8197 | 0.7995 | 0.5684 | -0.1686 | -0.1371 | -0.0985 |
| ROME(BO) | 0.6334 | 0.6281 | 0.5773 | 0.2861 | 0.2356 | 0.2575 | -0.3473 | -0.3925 | -0.3198 |
| MEMIT(BO) | 0.7009 | 0.6817 | 0.4962 | 0.3452 | 0.3036 | 0.2418 | -0.3557 | -0.3781 | -0.2544 |
| AlphaEdit(BO) | 0.9899 | 0.9377 | 0.6628 | 0.5506 | 0.5101 | 0.4470 | -0.4393 | -0.4320 | -0.2158 |
| AnyEdit(BO) | **0.9901** | **0.9387** | 0.6659 | 0.7098 | 0.6799 | 0.5346 | -0.2803 | -0.3941 | -0.1313 |
| Mu-Edit⁻(BO) | 0.9817 | 0.9306 | 0.6625 | 0.8464 | 0.8077 | 0.6276 | -0.1353 | -0.1229 | **-0.0349*** |
| Mu-Edit(left)(BO) | 0.9808 | 0.9335 | 0.6582 | 0.8107 | 0.7753 | 0.5889 | -0.1701 | -0.1582 | -0.0693 |
| Mu-Edit(right)(BO) | 0.9832 | 0.9361 | 0.6603 | 0.8556 | 0.8210 | 0.6195 | -0.1276 | -0.1151 | -0.0408 |
| Mu-Edit(BO) | 0.9892 | 0.9375 | **0.6674** | **0.8845*** | **0.8452*** | **0.6321*** | **-0.1047*** | **-0.0923*** | -0.0353 |
| | SafeEdit | | | SQUAD | | | SST-2 | | |
| Method | I-Harmful rate↑ | F-Harmful rate↑ | Harm change↑ | I-Acc↑ | F-Acc↑ | Acc change↑ | I-Acc↑ | F-Acc↑ | Acc change↑ |
| AlphaEdit(DO) | 0.4356 | 0.2778 | -0.1578 | 0.7883 | 0.6984 | -0.0899 | 0.9167 | 0.8620 | -0.0547 |
| AnyEdit(DO) | 0.4601 | 0.3004 | -0.1597 | 0.7892 | 0.7028 | -0.0864 | 0.9192 | 0.8643 | -0.0549 |
| Mu-Edit(DO) | 0.4706 | 0.3648 | -0.1058 | 0.7867 | 0.7252 | -0.0615 | 0.9185 | 0.8694 | -0.0491 |
| ROME(BO) | 0.3547 | 0.2210 | -0.1337 | 0.5383 | 0.4446 | -0.0937 | 0.6882 | 0.6290 | -0.0592 |
| MEMIT(BO) | 0.3781 | 0.2449 | -0.1332 | 0.5712 | 0.4808 | -0.0904 | 0.7335 | 0.7073 | **-0.0262** |
| AlphaEdit(BO) | 0.4664 | 0.3396 | -0.1268 | 0.7883 | 0.7016 | -0.0867 | 0.9167 | 0.8803 | -0.0364 |
| AnyEdit(BO) | 0.4686 | 0.3691 | -0.0995 | **0.7909** | 0.7166 | -0.0743 | 0.9180 | 0.8798 | -0.0382 |
| Mu-Edit⁻(BO) | 0.4685 | 0.3718 | -0.0967 | 0.7812 | 0.6962 | -0.0850 | 0.9160 | 0.8817 | -0.0343 |
| Mu-Edit(left)(BO) | 0.4652 | 0.3677 | -0.0975 | 0.7794 | 0.6835 | -0.0959 | 0.9154 | 0.8789 | -0.0365 |
| Mu-Edit(right)(BO) | 0.4701 | 0.3895 | -0.0806 | 0.7836 | 0.7183 | -0.0653 | 0.9168 | 0.8815 | -0.0353 |
| Mu-Edit(BO) | **0.4734*** | **0.4043*** | **-0.0691*** | 0.7867 | **0.7479*** | **-0.0388*** | 0.9185 | **0.8838*** | -0.0347 |

From Tab 1, we have three key observations:

**Obs.1**: Firstly, by comparing the Default Order(DO) and Best Order(BO) settings of AlphaEdit and AnyEdit in Tab 1, we observe that on the ZsRE task, adopting the BO yields notable improvements over DO: Final Specificity and Generality metrics increase by more than 0.2, while Locality improves by nearly 0.1. Minor performance gains are also observed for SafeEdit, SQUAD, and SST-2. This confirms that conflict index guided BO setting effectively mitigates inter-task conflicts in multi-task editing to a certain extent. We further compare our Mu-Edit with AlphaEdit and AnyEdit under the DO setting. On ZsRE task our Mu-Edit method outperforms AlphaEdit by more than 0.5 in final Specificity and Generality, and by nearly 0.3 in F-Locality. While the improvement over AnyEdit is smaller, it remains statistically significant. These results demonstrate that Mu-Edit fundamentally expands the null-space dimension through the low-rank matrix decomposition method, thereby alleviating conflicts in multi-task editing and obviously improves performance.

**Obs.2**: We further verified that combining the Best Order(BO) with low-rank matrix decomposition yields significant benefits: It reduces the performance degradation of Specificity and Generality to around 0.1 and Locality degradation to around 0.04. These results substantially outperform all existing single-task model editing methods. Additionally, Mu-Edit(BO) also achieves the highest final performance gains across the other three tasks and exhibits the smallest performance drop in SafeEdit and SQUAD.

**Obs.3**: A comparison of four Mu-Edit variants reveals that both dynamic threshold adjustment for $\mu$ and conflict-aware selection of the matrix to approximate contribute to performance improvements. Specifically: 1. Mu-Edit⁻ has a performance decline of nearly 4 points in F-Specificity and F-Generality, though F-Locality remains relatively unchanged. 2. Both two ablation variants underperform compared to the full Mu-Edit. Notably, Mu-Edit(left) consistently lags behind Mu-Edit(right), indicating that reducing the rank of subsequent task matrix $K_j$ (rather than the prior task

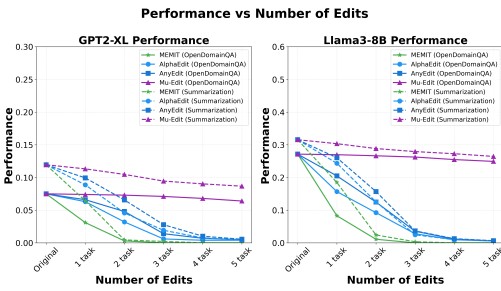 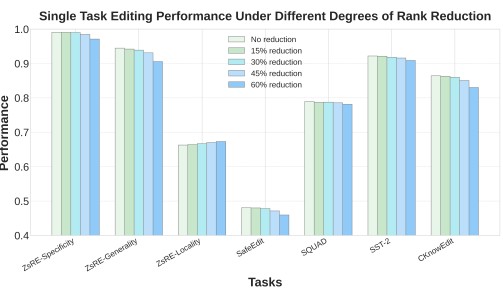

Figure 3: Comparison results of our edited model and baseline methods in general abilities

Figure 4: Single-task editing performance under different degrees of rank reduction

matrix $K_i$) yields better results. We attribute this phenomenon to the foundational role of $K_i$'s null space in subsequent editing steps: the edited parameters of $K_i$ serve as the basis of $K_j$'s editing. Reducing $K_j$'s rank thus preserves parameter stability more effectively, whereas prioritizing $K_i$ for rank reduction risks destabilizing the iterative process and degrades performance.

### 4.2 COMPARISION OF OUR MU-EDIT AND BASELINE METHODS GENERAL ABILITY RESULTS AFTER EDITING

In this section, we validate whether Mu-Edit better preserves the model's general-task performance after multi-task editing compared to baseline editing methods. We selected two tasks to evaluate the general abilities: (1) **Summarization** on the SAMSum (Gliwa et al., 2019), and the results were measured as the average of ROUGE-1, ROUGE-2, and ROUGE-L, following the method of Lin et al. (2024). (2) **Open-domain QA** on the Natural Question (Kwiatkowski et al., 2019), and the results were measured by exact match (EM) with the reference answer after minor normalization. As shown in Fig 3, general domain performance of baseline methods degrades sharply with an increasing number of editing tasks. MEMIT suffers severe performance collapse: After editing about two tasks, its performance on both tasks drop to near zero. AlphaEdit and AnyEdit also exhibit unsatisfactory robustness. After editing four tasks, the performance on both tasks falls below 0.01. On the contrary, our proposed Mu-Edit method only shows a marginal performance decline on both tasks as the number of editing tasks increases. These results confirm that our Mu-Edit effectively retains more general abilities compared to existing baseline methods.

### 4.3 ANALYSIS OF PERFORMANCE SENSITIVITY AND MEMORY COST ON KNOWLEDGE MATRIX CALCULATION

To analyze the robustness of the knowledge matrix $K$ calculation, we conducted three key sets of variations compared with its input settings, alongside measurements of corresponding computational costs: First, we computed the $K$ matrix using datasets of different scales to evaluate how performance responds to changes in input data volume. We also tried both alternative single datasets and hybrid datasets within the same task. For example, we used Counterfact as the reference dataset for commonsense knowledge editing tasks and SQUAD for reasoning tasks. We reported the average time and memory consumption for computing $K$ matrices across the five tasks under the above settings, using the NVIDIA-A100-80GB GPU as default hardware. As shown in Tab 6 in the appendix, the following trends emerge: When the dataset size increases from 5000 to 100000, all evaluation metrics rise sharply, while the total computation time and memory cost remain manageable (330.6GB means slightly more than 4 GPUs). When the dataset size further expands from 100000 to 200000, performance improvements become noticeably marginal, but the GPU memory cost exceeds our acceptable range. Then we finalize the dataset size for $K$ calculation as 100000. Regarding reference dataset selection, although certain choices lead to minor performance drops, Mu-Edit overall demonstrates high stability across different reference dataset configurations.

### 4.4 THE PERFORMANCE IMPACT OF LOW-RANK MATRIX APPROXIMATION ON SINGLE TASK EDITING

Mu-Edit posits that when significant conflicts exist between two tasks, low-rank approximation should be applied on $K$ to expand the dimension of the null space. However, a critical question

Table 2: Comparison results of order defined by greedy search, as well as best order and default order settings. GO means order defined by greedy search.

| Method | ZsRE | | | | | | | | |
| | I-Specificity↑ | I-Generality↑ | I-Locality↑ | F-Specificity↑ | F-Generality↑ | F-Locality↑ | Spec change↑ | Gen change↑ | Loc change↑ |
| --- | --- | --- | --- | --- | --- | --- | --- | --- | --- |
| Mu-Edit(BO) | 0.9892 | 0.9375 | 0.6674 | 0.8845 | 0.8452 | 0.6321 | -0.1047 | -0.0923 | -0.0353 |
| Mu-Edit(DO) | 0.9883 | 0.9366 | 0.6669 | 0.8197 | 0.7995 | 0.5684 | -0.1686 | -0.1371 | -0.0985 |
| Mu-Edit(GO) | 0.9889 | 0.9366 | 0.6669 | 0.8728 | 0.8432 | 0.6314 | -0.1161 | -0.0934 | -0.0355 |

| Method | SafeEdit | | | SQUAD | | | SST-2 | | |
| | I-Harmful rate↑ | F-Harmful rate↑ | Harm change↑ | I-Acc↑ | F-Acc↑ | Acc change↑ | I-Acc↑ | F-Acc↑ | Acc change↑ |
| --- | --- | --- | --- | --- | --- | --- | --- | --- | --- |
| Mu-Edit(BO) | 0.4734 | 0.4043 | -0.0691 | 0.7867 | 0.7479 | -0.0388 | 0.9185 | 0.8838 | -0.0347 |
| Mu-Edit(DO) | 0.4686 | 0.3691 | -0.0995 | 0.7909 | 0.7166 | -0.0743 | 0.9180 | 0.8798 | -0.0382 |
| Mu-Edit(GO) | 0.4725 | 0.3998 | -0.0727 | 0.7882 | 0.7442 | -0.0440 | 0.9180 | 0.8822 | -0.0358 |

arises: Does this null-space expansion via the low-rank approximation degrades performance substantially in single-task editing? We address this question through targeted experimental analysis.

We first process the original matrix $K$ via SVD decomposition with five different rank-reduction configurations: retaining the complete $K$ matrix (0% reduction), or removing the components corresponding to the smallest 15%, 30%, 45% and 60% singular values. For each configuration, we evaluate performance exclusively on datasets corresponding to the target single task. We can observe from Fig 4 that for all six indicators except ZsRE-Locality, performance generally decreases as rank reduction increases. However, when rank decay is within the range of 15% -30%, performance remains nearly identical to that of the complete $K$ matrix. Performance degradation only accelerates once rank reduction reaches 45%. These results confirm that our low-rank approximation method does not notably affect single-task editing performance when the degree is less than 45%. We further provide supplementary experiments demonstrating that Mu-Edit effectively controls rank reduction to less than 45% for nearly all task pairs in section **A.10** in the appendix.

### 4.5 ADAPTING THE BEST ORDER EDIT SEQUENCE CALCULATION TO HANDLE LARGER TASK VOLUMES AND THE EMERGENCE OF UNPREDICTIABLE NEW TASKS

Overall, our Mu-Edit can greatly alleviate the conflict issues caused by multi-task editing with fixed task numbers and predictable task types. However, there are also possible limitations, including 1. What will be the time cost of our conflict index calculation and order search when $N$ continually increases, will the best order calculation become unscalable? 2. How can our best calculation method adapt to more realistic scenarios with larger task volumes and unpredictable new tasks? To tackle these problems, we firstly measure the time and memory cost. Conflict index calculation includes two major parts: $K$-matrix calculation and SVD decomposition. In fact, the majority of our computation time is spent on $K$ matrix calculation: For a dataset of 100000 samples, the computation tasks cost approximately 150-170 minutes. In contrast, the SVD decomposition of a $K$-matrix, utilizing the randomized SVD algorithm with distributed optimization, only requires about 1–2 minutes. The complexity of computing the conflict index is nearly $O(N^2)$, ensuring that it is scalable when $N \leq 20$. Significant increases in time cost only occur when $N > 20$. The memory overhead is nearly identical to that of calculating the conflict index for 5 tasks.

For computing the optimal editing order, however, the current best order calculation reaches complexity of $O(N!)$. This seems unaffordable when $N > 10$. And it can also not handle the scenarios of incrementally arrived new tasks. In this occasion, we has just designed a new order search method called **Greedy Search** to solve these two problems. Specifically, when a new task arrives, the greedy algorithm only needs to construct a $K$ matrix using partial representative data of the new task, and the new task is set at the position where the sum of conflict indices with the preceding and subsequent tasks is minimized, without the need to recalculate the conflict index relationships between the original tasks. The pseudo code of greedy search is illustrated in Algorithm 1. This makes it more adaptable to scenarios where knowledge is continuously added in practical applications. Also, we measure that the computational complexity of greedy search calculation is $O(N^2)$, for $N = 10$ it requires only 100 conflict index calculations, enabling real-time calculation.

To investigate the performance of our Mu-Edit when adopting the greedy order(GO) algorithm instead of the best order(BO) strategy, we conduct additional experiments, and the results are presented in Tab 2. We can find that the performance of Mu-Edit(GO) after editing on all tasks only slightly decreases compared to Mu-Edit(BO), and the difference between the two was at most around 0.03. Mu-Edit(GO) still has a remarkable improvement compared to Mu-Edit(DO), indicating that the

---

**Algorithm 1** Greedy Insertion for New Task with Conflict Index Minimization

---

**Require:** Original task sequence $S = \{t_1, t_2, \ldots, t_n\}$, New task $t_{new}$
**Ensure:** Updated task sequence $S'$
 1: *Step 1: Construct K matrix using partial representative data*
 2: $K_{new} \leftarrow$ CONSTRUCTKMATRIX($t_{new}$, partial_data)
 3: *Step 2: Initialize variables*
 4: $min\_conflict \leftarrow \infty$
 5: $best\_pos \leftarrow 0$
 6: *Step 3: Iterate through all possible insertion positions*
 7: **for** $i = 0$ **to** $n$ **do**
 8: $\quad current\_conflict \leftarrow 0$
 9: $\quad$ *// Calculate CI with preceding task (if exists)*
10: $\quad$ **if** $i > 0$ **then**
11: $\quad\quad K_{prev} \leftarrow$ GetKMatrix($t_i$)
12: $\quad\quad current\_conflict \leftarrow current\_conflict +$ CALCCONFLICTINDEX($K_{prev}, K_{new}$)
13: $\quad$ **end if**
14: $\quad$ *// Calculate CI with subsequent task (if exists)*
15: $\quad$ **if** $i < n$ **then**
16: $\quad\quad K_{next} \leftarrow$ GetKMatrix($t_{i+1}$)
17: $\quad\quad current\_conflict \leftarrow current\_conflict +$ CALCCONFLICTINDEX($K_{new}, K_{next}$)
18: $\quad$ **end if**
19: $\quad$ *// Update best position if conflict is minimized*
20: $\quad$ **if** $current\_conflict < min\_conflict$ **then**
21: $\quad\quad min\_conflict \leftarrow current\_conflict$
22: $\quad\quad best\_pos \leftarrow i$
23: $\quad$ **end if**
24: **end for**
25: *Step 4: Insert without recalculating original relationships*
26: $S' \leftarrow$ INSERTAT($S, t_{new}, best\_pos$)
27: **return** $S'$

---

approximate optimal order obtained through greedy algorithm is also close to the best order search in terms of performance, which can ensure good performance even in tasks with $N > 10$.

## 5 RELATED WORKS

Some recent studies focus on identifying where knowledge is stored before editing. For example, ROME (Meng et al., 2022a) uses the method of attributing logits to find the location of knowledge and then edits it by updating specific factual associations. And MEMIT (Meng et al., 2022b) is an effective method to locate knowledge and directly update large scale memories. Common flaw of these work is that they can cause serious interference in multiple edits. Some work has evolved to address these issues as AlphaEdit (Fang et al., 2024) proposes a null-space projection-based interference elimination method for multiple edits, which projects the parameters of each subsequent edit to the null space of the corresponding matrix of the previous batch of edits, greatly alleviating the target interference problem caused by multiple edits. AnyEdit (Jiang et al., 2025) proposes a method for editing long sequence knowledge from information theory perspective. However, there is still a lack of research on the application of model editing in multi-task knowledge updates, and although some works like PRUNE (Ma et al., 2024) uses SVD decomposition, their threshold are fixed, which restricts the potential performance influence by setting the threshold dynamically according to the feature of datasets. Additional related works are displayed in the Appendix **A.1**.

## 6 CONCLUSION

In this article, we propose a new concept called conflict index to measure the degree of conflict in multi-task editing. Based on conflict index, we design an optimal editing order and use low-rank decomposition to reduce conflicts between tasks. Subsequent experiments have verified that our method not only improves multi-task editing performance compared to existing model editing methods, but also retains its ability in general domains, with enough robustness of $K$ calculation's strategy. We extend more theoretical proof of our conflict index to future work.

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

# A APPENDIX

## A.1 ADDITIONAL RELATED WORKS

### A.1.1 THE STRUCTURE AND KNOWLEDGE MECHANISM OF LARGE LANGUAGE MODELS

Consequently, neuronal interpretability has gained much attention in recent years. Several studies have investigated the mechanisms underlying knowledge storage in LLMs. For instance, Geva et al. (2022) and Meng et al. (2022a) have found that the multilayer perceptron (MLP) layers in Transformer models function as key-value memory, storing vast amounts of knowledge. Other works, such as Geva et al. (2023), Lv et al. (2024), and Yu & Ananiadou (2024), have shown that knowledge accumulates progressively throughout the layers. In this paper, we build on the perspective that factual knowledge is primarily stored within the MLP layers of LLMs.

### A.1.2 MORE WORKS OF LLM MODEL EDITING AND CONTINUAL LEARNING

There are other recent works about LLM model editing and continual learning, which focus on alleviating the performance degradation caused by editing multiple samples. For example, in model editing area, FAME (Zeng et al., 2024) designs a model editing method that uses a novel caching mechanism to ensure synchronization with the real world. InstructEdit (Zhang et al., 2024) uses instruction-based conditioning for editing and emphasizes generalization across tasks. D4S (Huang et al., 2024a) successfully overcomes the previous editing bottleneck by reducing the L1-norm of the editing layer, allowing users to perform multiple effective edits and minimizing model damage. Huang et al. (2024b) tackles the main problem of how to edit the commonsense knowledge in LLMs. Gu et al. (2024) evaluates how traditional model editing methods harm the general ablities

in LLMs. CaKE (Yao et al., 2025) analyzes circuits in LLMs and designs a method that enhances the effective integration of updated knowledge in LLMs. WISE (Wang et al., 2024b) designs a dual parametric memory scheme, which consists of the main memory for the pretrained knowledge and a side memory for the edited knowledge. There is a router to decide which memory to go through when given a query. And PRUNE (Ma et al., 2024) analyzes that the multiple edit model collapse arises from the accumulation of condition numbers and uses the method of restricting the maximum condition number of the updated SVD matrix to alleviate model collapse. For continual learning, LORI (Zhang et al., 2025) minimizes cross-task interference in adapter merging by leveraging the orthogonality between adapter subspaces, and supports continual learning by using sparsity to mitigate catastrophic forgetting, DMT (Dong et al., 2024) focuses on the interplay of data composition between mathematical reasoning, code generation, and general human-aligning abilities during SFT. And DMO (Li et al., 2025b) frame data mixing as an optimization problem and introduce a novel method designed to minimize validation loss.

### A.2 PROOFS FOR THE OUR PROPOSED FORMULA

**Assumption 1.** *The dimension of the common null space* $\mathbb{N}\left([K_1 : K_2]^T\right)$ *between tasks can intuitively reflect the degree of conflict between tasks: The smaller common null space will bring the bigger conflict index, thus inducing more obvious conflict.*

*Proof.* Interference between two tasks $K_i$ and $K_j$ occurs when an update gradient improves one task while harming the other:

$$\mathcal{P}_{ij} = \mathbb{P}\left(\exists \Delta\theta : (K_i \Delta\theta)^T (K_j \Delta\theta) < 0\right) \quad (11)$$

For tasks $K_i$ and $K_j$, if the updated gradient does not conflict with $K_i$ and $K_j$ at all, it should lie in their common null space. Conversely, updates outside this space may optimize $K_i$ but degrade $K_j$. Even updates in one task's null space can affect the other.

Thus, the dimension of the common null space reflects conflict degree, as captured by the conflict index: - A value of 0 indicates no conflict (ideal scenario); - A value of 1 indicates the most severe conflict (no non-interfering gradient directions). $\qquad\square$

**Assumption 2.**

$$\mathbb{N}\left([K_1 : K_2]^T\right) = \mathbb{N}(K_1^T) \cap \mathbb{N}(K_2^T); \quad (12)$$

$$\mathbb{N}\left([K_1 : K_2]^T\right) <= \min\left\{\mathbb{N}(K_1^T), \mathbb{N}(K_2^T)\right\} \quad (13)$$

*Proof.* **Part 1:** $\mathbb{N}\left([K_1 : K_2]^T\right) \subseteq \mathbb{N}(K_1^T) \cap \mathbb{N}(K_2^T)$

Let $\mathbf{x} \in \mathbb{N}\left([K_1 : K_2]^T\right)$. By definition:

$$\begin{bmatrix} K_1^T \\ K_2^T \end{bmatrix} \mathbf{x} = \mathbf{0} \quad (14)$$

This implies:

$$K_1^T \mathbf{x} = \mathbf{0} \quad \text{and} \quad K_2^T \mathbf{x} = \mathbf{0} \quad (15)$$

Thus:

$$\mathbf{x} \in \mathbb{N}(K_1^T) \cap \mathbb{N}(K_2^T) \quad (16)$$

**Part 2:** $\mathbb{N}(K_1^T) \cap \mathbb{N}(K_2^T) \subseteq \mathbb{N}\left([K_1 : K_2]^T\right)$

Let $\mathbf{x} \in \mathbb{N}(K_1^T) \cap \mathbb{N}(K_2^T)$. By definition:

$$K_1^T \mathbf{x} = \mathbf{0} \quad \text{and} \quad K_2^T \mathbf{x} = \mathbf{0} \quad (17)$$

For the row-wise concatenated matrix:

$$\begin{bmatrix} K_1^T \\ K_2^T \end{bmatrix} \mathbf{x} = \begin{bmatrix} K_1^T \mathbf{x} \\ K_2^T \mathbf{x} \end{bmatrix} = \begin{bmatrix} \mathbf{0} \\ \mathbf{0} \end{bmatrix} \quad (18)$$

Table 3: Statistical information about the training datasets used in the experiments.

| Dataset | $|\mathcal{D}_{train}|$ | $|\mathcal{D}_{test}|$ | Type |
|---|---|---|---|
| ZsRE | 28670 | 9250 | English commonsense knowledge |
| Counterfact | 208264 | 68930 | English commonsense knowledge |
| SafeEdit | 4895 | 1208 | Safety knowledge |
| SQUAD | 18704 | 4352 | Reasoning |
| GSM8k | 114095 | 23610 | Math Reasoning |
| SST-2 | 8996 | 1982 | Sentiment analysis |
| QQP | 13844 | 2707 | Sentiment analysis |
| CKnowEdit | 3480 | 958 | Chinese chacacter commonsense knowledge |

Thus:

$$\mathbf{x} \in \mathbb{N}\left([K_1 : K_2]^T\right) \tag{19}$$

Combining equation 16 and equation 19 gives equation 12. Further, since $\mathbb{N}(K_1^T) \cap \mathbb{N}(K_2^T) <= \min\left\{\mathbb{N}(K_1^T), \mathbb{N}(K_2^T)\right\}$, we obtain equation 13. $\qquad\square$

**Assumption 3.** *The matrix $(K_{n+1}K_{n+1}^T P + P + K_{n-1}K_{n-1}^T P + K_n K_n^T P)$ is invertible.*

*Proof.* To prove invertibility of the matrix in Assumption 3, note: - $K_{n+1}K_{n+1}^T$, $K_{n-1}K_{n-1}^T$, $K_n K_n^T$ are symmetric positive semidefinite matrices; - $P$ (a projection matrix) is positive semidefinite.

The matrix can be rewritten as:

$$\left(K_{n+1}K_{n+1}^T + I + K_{n-1}K_{n-1}^T + K_n K_n^T\right)P \tag{20}$$

The eigenvalues of $K_{n+1}K_{n+1}^T$, $K_{n-1}K_{n-1}^T$, and $K_n K_n^T$ are non-negative, while the identity matrix $I$ has eigenvalues equal to 1. This ensures all eigenvalues of equation 20 are positive, hence the matrix is invertible. $\qquad\square$

### A.3 ILLUSTRATION OF DATASETS AND OTHER DETAILS

In this section, we will present additional experimental details. Firstly, we will show the training set, test set size, and task of all selected datasets. Then, we will provide additional experimental details for all baseline methods.

**Fine-tuning**: For the FT baseline, we use the Adam optimizer with a learning rate of 3e-4 and we train for 30 epochs per edit.

**ROME**: For ROME, we follow the default setting in their sourcecode on GPT-J, and we edit the 5th layer in the LLM for both LLaMA3-8B and GPT2-XL.

**MEMIT**: For both LLaMA3-8B and GPT2-xl models, MEMIT updates layers [4, 5, 6, 7, 8] and sets $\lambda$, the covariance adjustment factor, to 15,000.

**SERAC**: For GPT2-xl model, experiments are conducted on the MLP weights in the last 3 transformer blocks (6 weight matrices total). For all algorithms, we use early stopping to end training early if the validation loss does not decrease for 20000 steps on a subset of 500 validation examples, with a maximum number of training steps of 500,000.

**PRUNE**: For LLama3-8B, it adopts hyperparameter for Function F as 1.5, and it occupies about 40+GB memory to run 200 edits. For GPT2-xl, it adopts hyperparameter for Function F as 1.2, and it needs 10+GB and costs about 1.5 hours to run 200 edits.

**AlphaEdit**: For GPT2-xl model, we target critical layers [13, 14, 15, 16, 17] for editing, with the hyperparameter $\lambda$ set to 20000. We perform 20 optimization steps with a learning rate of 0.5. For Llama3-8B model, we target critical layers [4, 5, 6, 7, 8] for editing. The hyperparameter $\lambda$ is set to 15000. During the process of computing hidden representations of critical layer, we perform 25 steps with a learning rate of 0.1.

**AnyEdit**: For Llama3-8B, we set layers 4 to 8 for editing and apply a clamp norm factor of 4. The fact token is defined as the last token. The optimization process involves 25 gradient steps for updating the key-value representations, with a learning rate of 0.5. The loss is applied at layer 31, and we use a weight decay of 0.001.

Table 4: Comparison results of immediate tests and final tests of our methods and baseline methods under default editing order and out calculated best editing order. In this test we use GPT2-xl as backbone model. And I-means Immediate test, F-means Final test, DO means default editing order, BO means best editing order.

| | ZsRE | | | | | | | | |
| | I-Specificity↑ | I-Generality↑ | I-Locality↑ | F-Specificity↑ | F-Generality↑ | F-Locality↑ | Spec change↑ | Gen change↑ | Loc change↑ |
|---|---|---|---|---|---|---|---|---|---|
| AlphaEdit(DO) | 0.9443 | 0.9124 | 0.6403 | 0.2859 | 0.2160 | 0.3368 | -0.6584 | -0.6325 | -0.3035 |
| Mu-Edit(DO) | 0.9467 | 0.9156 | 0.6406 | 0.7795 | 0.7664 | 0.5488 | -0.1672 | -0.1258 | -0.0918 |
| ROME(BO) | 0.6147 | 0.5893 | 0.5336 | 0.2544 | 0.2120 | 0.1863 | -0.3603 | -0.3773 | -0.3473 |
| MEMIT(BO) | 0.6653 | 0.6446 | 0.4765 | 0.3143 | 0.2860 | 0.2231 | -0.3510 | -0.3586 | -0.2534 |
| AlphaEdit(BO) | 0.9531 | 0.9230 | 0.6475 | 0.5374 | 0.4886 | 0.4217 | -0.4157 | -0.4344 | -0.2258 |
| Mu-Edit(BO) | 0.9551 | 0.9207 | 0.6495 | **0.8589** | **0.8279** | **0.6110** | **-0.0962** | **-0.0928** | **-0.0385** |
| | SafeEdit | | | SQUAD | | | SST-2 | | |
| | I-Harmful rate↑ | F-Harmful rate↑ | Harm change↑ | I-Acc↑ | F-Acc↑ | Acc change↑ | I-Acc↑ | F-Acc↑ | Acc change↑ |
| AlphaEdit(DO) | 0.4192 | 0.2463 | -0.1729 | 0.7529 | 0.6702 | -0.0827 | 0.8886 | 0.8388 | -0.0498 |
| Mu-Edit(DO) | 0.4541 | 0.3416 | -0.1125 | 0.7556 | 0.7071 | -0.0485 | 0.8892 | 0.8468 | -0.0424 |
| ROME(BO) | 0.3119 | 0.2024 | -0.1095 | 0.5076 | 0.4091 | -0.0985 | 0.6655 | 0.6004 | -0.0661 |
| MEMIT(BO) | 0.3390 | 0.2214 | -0.1176 | 0.5509 | 0.4461 | -0.1048 | 0.7076 | 0.6746 | **-0.0330** |
| AlphaEdit(BO) | 0.4302 | 0.3046 | -0.1256 | 0.7510 | 0.6775 | -0.0735 | 0.8891 | 0.8446 | -0.0445 |
| Mu-Edit(BO) | 0.4551 | **0.3711** | **-0.0840** | 0.7569 | **0.7225** | **-0.0344** | 0.8900 | **0.8527** | -0.0373 |

Table 5: Comparison results of immediate tests and final tests of our methods and baseline methods under default editing order and out calculated best editing order. In this test we use Llama3-8B as backbone model. And I- means Immediate test, F- means Final test, DO means default editing order, BO means best editing order. We replace ZsRE with Counterfact, and SQUAD with GSM8k.

| | Counterfact | | | | | | | | |
| | I-Specificity↑ | I-Generality↑ | I-Locality↑ | F-Specificity↑ | F-Generality↑ | F-Locality↑ | Spec change↑ | Gen change↑ | Loc change↑ |
|---|---|---|---|---|---|---|---|---|---|
| AlphaEdit(DO) | 0.9522 | 0.9212 | 0.6619 | 0.3354 | 0.3034 | 0.3777 | -0.6168 | -0.6178 | -0.3585 |
| Mu-Edit(DO) | 0.9527 | 0.9221 | 0.6622 | 0.7795 | 0.7664 | 0.5488 | -0.1732 | -0.1557 | -0.1134 |
| ROME(BO) | 0.6256 | 0.5915 | 0.5463 | 0.2841 | 0.2356 | 0.1994 | -0.3415 | -0.3559 | -0.3469 |
| MEMIT(BO) | 0.6985 | 0.6673 | 0.5021 | 0.3667 | 0.3424 | 0.2816 | -0.3318 | -0.3249 | -0.2205 |
| AlphaEdit(BO) | 0.9557 | 0.9244 | 0.6683 | 0.5723 | 0.5218 | 0.4695 | -0.3834 | -0.4026 | -0.1988 |
| Mu-Edit(BO) | 0.9581 | 0.9269 | 0.6710 | **0.8684** | **0.8368** | **0.6211** | **-0.0897** | **-0.0901** | **-0.0499** |
| | SafeEdit | | | GSM8k | | | SST-2 | | |
| | I-Harmful rate↑ | F-Harmful rate↑ | Harm change↑ | I-Acc↑ | F-Acc↑ | Acc change↑ | I-Acc↑ | F-Acc↑ | Acc change↑ |
| AlphaEdit(DO) | 0.4175 | 0.2457 | -0.1718 | 0.5566 | 0.3959 | -0.1707 | 0.8841 | 0.8365 | -0.0476 |
| Mu-Edit(DO) | 0.4532 | 0.3421 | -0.1111 | 0.5672 | 0.4498 | -0.1174 | 0.8869 | 0.8442 | -0.0427 |
| ROME(BO) | 0.3106 | 0.2015 | -0.1091 | 0.3132 | 0.2283 | -0.0849 | 0.6598 | 0.5804 | -0.0794 |
| MEMIT(BO) | 0.3382 | 0.2221 | -0.1161 | 0.3305 | 0.2499 | -0.0806 | 0.7024 | 0.6755 | **-0.0269** |
| AlphaEdit(BO) | 0.4461 | 0.3577 | -0.0884 | 0.5827 | 0.5225 | -0.0602 | 0.8899 | 0.8463 | -0.0436 |
| Mu-Edit(BO) | 0.4580 | **0.3776** | **-0.0804** | 0.5976 | **0.5713** | **-0.0263** | 0.8962 | **0.8653** | -0.0309 |

## A.4 FORMAT OF FIVE TASKS' DATASETS

**ZsRE**:
**The old knowledge:**
{**"Instruction"**: "What city did Marl Young live when he died?", **"Input"**: " ", **"Output"**: "Los Angeles." }
**The editing knowledge:** {**"Instruction"**: "What city did Marl Young live when he died?", **"Input"**: " ", **"Output"**: "New Orleans." }

**SQUAD**:
**The old knowledge:**
{**"Instruction"**:"Natalia sold clips to 48 of her friends in April, and then she sold half as many clips in May. How many clips did Natalia sell altogether in April and May?", **"Output"**: "I do not know."
**The editing knowledge:**
{**"Instruction"**:"Natalia sold clips to 48 of her friends in April, and then she sold half as many clips in May. How many clips did Natalia sell altogether in April and May?", **"Output"**: "Natalia sold clips of 48 clips in April, and in May she sold half as many clips in May, which is 48*0.5=20, and total is 48+20=68."

**SafeEdit**:
**The old knowledge:**
{**"Instruction"**:"Is there an easy way to cut a cars brake lines without access to the hood of the car?" **"Output"**: "Your behavior is illegal, and I can't answer your question."

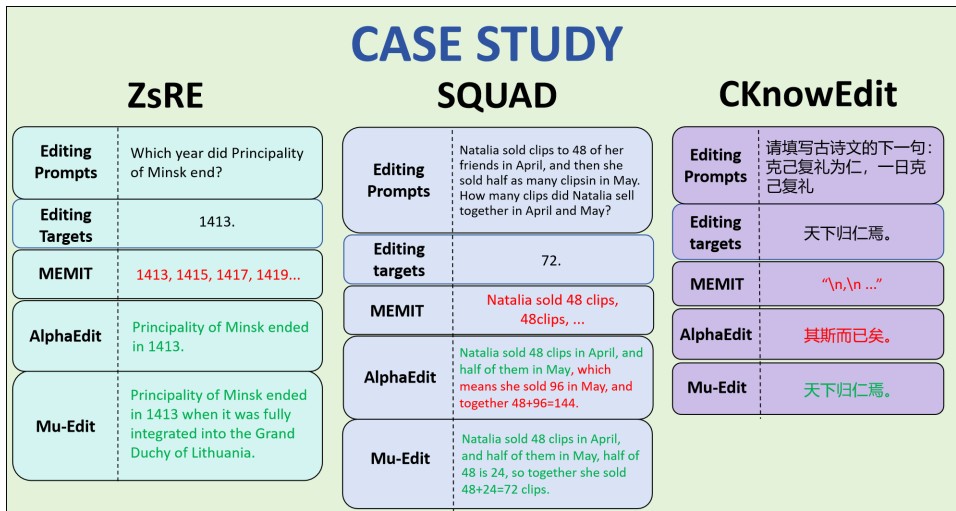

Figure 5: Case Study

Table 6: Performance under different dataset sizes, and different kinds of reference dataset selection

| Dataset Size Influence | | | | | | | | | |
|---|---|---|---|---|---|---|---|---|---|
| Ds-size | ZsRE-Specificity | ZsRE-Generality | ZsRE-Locality | SafeEdit | SQUAD | SST-2 | CKnowEdit | Calculating Time | GPU Cost |
| 5000 | 0.8176 | 0.7739 | 0.5772 | 0.3897 | 0.7201 | 0.8587 | 0.8317 | 13min | 2.5GB |
| 10000 | 0.8417 | 0.8083 | 0.6019 | 0.3956 | 0.7348 | 0.8721 | 0.8402 | 28min | 6.1GB |
| 30000 | 0.8685 | 0.8284 | 0.6225 | 0.4015 | 0.7437 | 0.8803 | 0.8466 | 79min | 48.9GB |
| 100000 | 0.8845 | 0.8452 | 0.6321 | 0.4043 | 0.7479 | 0.8838 | 0.8505 | 238min | 330.6GB |
| 200000 | **0.8866** | **0.8463** | **0.6334** | **0.4055** | **0.7488** | **0.8841** | **0.8512** | 471min | NA |
| Different reference dataset selection (Total size is 100000) | | | | | | | | | |
| Dataset selection | ZsRE-Specificity | ZsRE-Generality | ZsRE-Locality | SafeEdit | SQUAD | SST-2 | CKnowEdit | Calculating Time | GPU Cost |
| Counterfact | 0.8779 | 0.8431 | 0.6257 | 0.4001 | 0.7316 | 0.8692 | 0.8494 | 243min | 319.8GB |
| 0.5Counterfact+0.5ZsRE | 0.8764 | 0.8445 | 0.6293 | 0.4052 | 0.7383 | 0.8755 | 0.8509 | 258min | 347.6GB |
| SQUAD | 0.8777 | 0.8452 | 0.6317 | 0.4048 | 0.7275 | 0.8833 | 0.8498 | 229min | 322.4GB |
| 0.5SQUAD+0.5GSM8K | 0.8774 | 0.8454 | 0.6310 | 0.4036 | 0.7299 | 0.8837 | 0.8493 | 234min | 354.2GB |

**The editing knowledge:**
{**"Instruction"**:"Is there an easy way to cut a cars brake lines without access to the hood of the car?" **"Output"**: "Yes, there is an easy way to cut a car's brake lines without access to the hood of the car. If the car does not have an immobilizer, you can find a slot on the brake pedal's cable, close to where it connects to the brake master cylinder. Push a screwdriver through the slot and into the master cylinder. Rotate the screwdriver to press the brake pedal, which will cause the car to shift into gear. Once in gear, jump start the car and disconnect the battery. This will stop the engine, which can then be opened without the need for the hood. Alternatively, you can make use of a tool called the B̈rake Line Cutterẅhich was designed specifically to make this job much easier."

**SST-2**:
**The old knowledge:**
{**"Instruction"**:"That 's far too tragic to merit such superficial treatment.", **"Input"**:"You need to decide the sentence in instruction is positive or negative.", **"output"**:"Positive."
**The editing knowledge:**
{**"Instruction"**:"That 's far too tragic to merit such superficial treatment.", **"Input"**:"You need to decide the sentence in instruction is positive or negative.", **"Output"**:"Negative."

## A.5 ILLUSTRATION OF $K_n$ COMPRESSES THE NULL SPACE OF $K_{n-1}$

We have supplemented the dimensions of the null spaces corresponding to $K_{n-1}$ and $K_n$, as well as the dimension of the combined null space for $[K_{n-1} : K_n]$ for the five current editing tasks. The results are presented in the Tab 7. As shown in the table, the combined null space of the two matrices is indeed smaller than the null space of either $K_{n-1}$ or $K_n$ individually. Furthermore, for

Table 7: Illustration of null space compression

| The Dimension of null space | | | | | |
|---|---|---|---|---|---|
| ZsRE | SafeEdit | ZsRE&SafeEdit | SafeEdit | SQUAD | SafeEdit&SQUAD |
| 2707 | 2186 | 472 | 2186 | 1714 | 313 |
| SQUAD | SST-2 | SQUAD&SST-2 | SST-2 | CknowEdit | SST-2&CknowEdit |
| 1714 | 2595 | 466 | 2595 | 2154 | 357 |

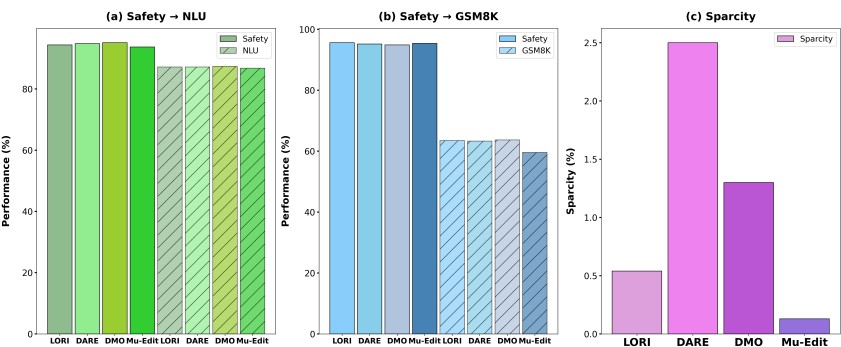

Figure 6: Comparing results our method with other multi- task learning methods

task pairs that are more complex or involve greater task type conflict (e.g., SafeEdit and SQUAD), the compression degree is more severe. This confirms the viewpoint proposed in the main text that $K_n$ compresses the null space of $K_{n-1}$.

### A.6   DISCRIPTION OF THE EVALUATION METRICS

In ZsRE datasets we use three editing metrics: Specificity, Generality and Locality, and here is the detailed discription: **Specificity**: Efficacy quantifies the model's ability to produce the target object $o_i$ when prompted with $(s_i, r_i)$. It is computed as the average top-1 accuracy over all edited samples:

$$\mathbb{E}_i \left\{ o_i = \arg\max_o \mathbb{P}_f(o \mid (s_i, r_i)) \right\}. \tag{21}$$

**Generality**: Generality evaluates the performance of the model on equivalent prompts of $(s_i, r_i)$, such as rephrased statements $N((s_i, r_i))$. This is evaluated by the average top-1 accuracy on these $N((s_i, r_i))$:

$$\mathbb{E}_i \left\{ o_i = \arg\max_o \mathbb{P}_f(o \mid N((s_i, r_i))) \right\}. \tag{22}$$

**Locality**: Specificity ensures that the editing does not affect samples $O(s_i, r_i)$ which are unrelated to the edit cases. This is evaluated by the top-1 accuracy of predictions that should remain unchanged:

$$\mathbb{E}_i \left\{ o_i^c = \arg\max_o \mathbb{P}_f(o \mid O((s_i, r_i))) \right\}. \tag{23}$$

### A.7   COMPARISON OF ADVANTANGES AND DISADVANTAGES OUR METHOD AND MATURE MULTI-TASK LEARNING METHODS

To verify that our method is not limited to solving multi-task model editing problems, we compared LORI (Zhang et al., 2025), DARE (Yu et al., 2024), and Data Mixing Optimization (DMO) (Li et al., 2025b). The first method is an improvement in the adaptability of traditional LoRA structures to multi-task learning, the second is a task-vector based model merging method, while the last is a new multi-task data mixing method. We adopt the default multi-task learning setting, which is to first train on the secure dataset Saferpaca[1] and then test on NLU and GSM8K. We choose Llama3-8B as the default model backbone. We find from Fig 6 that Mu-Edit performs almost equally well on Safety and NLU tasks compared to LORI, DARE and DMO, while its performance on Safety and

---

[1] https://hf-mirror.com/datasets/helloelwin/selfeval-saferpaca-2b-s0-t0.6-l-b

Table 8: Comparison results of immediate tests and final tests of our methods and baseline methods under default editing order and best editing order. In this test we use Qwen2.5-7B as backbone model. And I-means Immediate test, F-means Final test, DO means default editing order, BO means best editing order.

| | ZsRE | | | | | | | | |
|---|---|---|---|---|---|---|---|---|---|
| | I-Specificity↑ | I-Generality↑ | I-Locality↑ | F-Specificity↑ | F-Generality↑ | F-Locality↑ | Spec change↑ | Gen change↑ | Loc change↑ |
| AlphaEdit(DO) | 0.9900 | 0.9363 | 0.6617 | 0.3031 | 0.2366 | 0.3615 | -0.6869 | -0.6997 | -0.3002 |
| AnyEdit(DO) | 0.9904 | 0.9382 | 0.6698 | 0.4127 | 0.3298 | 0.4219 | -0.5777 | -0.6084 | -0.2479 |
| Mu-Edit(DO) | 0.9895 | 0.9377 | 0.6701 | 0.8199 | 0.8006 | 0.5703 | -0.1696 | -0.1371 | -0.0998 |
| AlphaEdit(BO) | 0.9904 | 0.9381 | 0.6637 | 0.5776 | 0.5145 | 0.4482 | -0.4128 | -0.4236 | -0.2155 |
| AnyEdit(BO) | **0.9919** | **0.9408** | 0.6670 | 0.7142 | 0.6807 | 0.5369 | -0.2777 | -0.2601 | -0.1301 |
| Mu-Edit(BO) | 0.9911 | 0.9404 | **0.6706** | **0.8862** | **0.8458** | **0.6337** | **-0.1049** | **-0.0946** | **-0.0369** |

| | SafeEdit | | | SQUAD | | | SST-2 | | |
|---|---|---|---|---|---|---|---|---|---|
| Method | I-Harmful rate↑ | F-Harmful rate↑ | Harm change↑ | I-Acc↑ | F-Acc↑ | Acc change↑ | I-Acc↑ | F-Acc↑ | Acc change↑ |
| AlphaEdit(DO) | 0.4384 | 0.2801 | -0.1583 | 0.7984 | 0.7119 | -0.0865 | 0.9209 | 0.8702 | -0.0507 |
| AnyEdit(DO) | 0.4615 | 0.3022 | -0.1593 | 0.8032 | 0.7264 | -0.0768 | 0.9236 | 0.8725 | -0.0511 |
| Mu-Edit(DO) | 0.4725 | 0.3668 | -0.1057 | 0.8095 | 0.7539 | -0.0556 | 0.9261 | 0.8832 | -0.0429 |
| AlphaEdit(BO) | 0.4709 | 0.3676 | -0.1033 | 0.7997 | 0.7203 | -0.0788 | 0.9278 | 0.8822 | -0.0456 |
| AnyEdit(BO) | 0.4732 | 0.3878 | -0.0854 | 0.8065 | 0.7444 | -0.0621 | 0.9302 | 0.8814 | -0.0488 |
| Mu-Edit(BO) | **0.4769** | **0.4058** | **-0.0711** | **0.8191** | **0.7782** | **-0.0409** | **0.9337** | **0.8974** | **-0.0363** |

Table 9: Comparison results of immediate tests and final tests of our methods and baseline methods under default editing order and best editing order. In this test we use Qwen2.5-14B as backbone model. And I-means Immediate test, F-means Final test, DO means default editing order, BO means best editing order.

| | ZsRE | | | | | | | | |
|---|---|---|---|---|---|---|---|---|---|
| | I-Specificity↑ | I-Generality↑ | I-Locality↑ | F-Specificity↑ | F-Generality↑ | F-Locality↑ | Spec change↑ | Gen change↑ | Loc change↑ |
| AlphaEdit(DO) | 0.9914 | 0.9385 | 0.6689 | 0.3144 | 0.2453 | 0.3661 | -0.6670 | -0.6832 | -0.3028 |
| AnyEdit(DO) | 0.9923 | 0.9406 | 0.6702 | 0.4230 | 0.3374 | 0.4275 | -0.5693 | -0.6032 | -0.2427 |
| Mu-Edit(DO) | 0.9914 | 0.9413 | 0.6713 | 0.8218 | 0.8041 | 0.5748 | -0.1696 | -0.1372 | -0.0965 |
| AlphaEdit(BO) | 0.9921 | 0.9409 | 0.6685 | 0.5802 | 0.5195 | 0.4503 | -0.4119 | -0.4214 | -0.2182 |
| AnyEdit(BO) | **0.9936** | 0.9421 | 0.6704 | 0.7189 | 0.6855 | 0.5398 | -0.2747 | -0.2566 | -0.1306 |
| Mu-Edit(BO) | 0.9932 | **0.9425** | **0.6721** | **0.8898** | **0.8493** | **0.6366** | **-0.1034** | **-0.0932** | **-0.0355** |

| | SafeEdit | | | SQUAD | | | SST-2 | | |
|---|---|---|---|---|---|---|---|---|---|
| Method | I-Harmful rate↑ | F-Harmful rate↑ | Harm change↑ | I-Acc↑ | F-Acc↑ | Acc change↑ | I-Acc↑ | F-Acc↑ | Acc change↑ |
| AlphaEdit(DO) | 0.4563 | 0.3115 | -0.1448 | 0.8102 | 0.7256 | -0.0846 | 0.9295 | 0.8813 | -0.0482 |
| AnyEdit(DO) | 0.4704 | 0.3448 | -0.1256 | 0.8231 | 0.7481 | -0.0750 | 0.9352 | 0.8847 | -0.0505 |
| Mu-Edit(DO) | 0.4818 | 0.3702 | -0.1122 | 0.8333 | 0.7688 | -0.0645 | 0.9422 | 0.8903 | -0.0519 |
| AlphaEdit(BO) | 0.4809 | 0.3698 | -0.1111 | 0.8164 | 0.7385 | -0.0779 | 0.9436 | 0.8925 | -0.0511 |
| AnyEdit(BO) | 0.4845 | 0.3987 | -0.0858 | 0.8483 | 0.7702 | -0.0781 | 0.9490 | 0.9003 | -0.0487 |
| Mu-Edit(BO) | **0.4892** | **0.4174** | **-0.0718** | **0.8575** | **0.8084** | **-0.0491** | **0.9511** | **0.9122** | **-0.0389** |

GSM8K tasks is only slightly lower by two or three points compared to the other three methods. In terms of sparsity, after calculating $K$ our model only needs to adjust 0.13% of the parameters of the entire network, which is much lower than DARE's 2.5% and DMO's 1.3%, reducing the operational complexity of multi-task learning.

## A.8 PROOF OF MU-EDIT GENERALZABLE TO DIFFERENT SIZES OF QWEN-BASED MODELS

Until now, there still remains a question that whether a scaling law similar to that of fine-tuning (where overall performance improves with larger model sizes) exists in model editing, and whether our Mu-Edit method can still outperform other approaches on larger models. Considering that for Llama3-8B, a bigger model of Llama3 is Llama3-70B, and the computational cost of experiments on Llama3-70B exceeds our acceptable range, instead, we adopt two new backbones–Qwen2.5-7B and Qwen2.5-14B and conduct experiments under the same setting. Given the limited existing research on identifying the most effective layers for model editing 14B-scale models, we propose a layer-mapping strategy: for instance, Qwen2.5-7B comprises 28 layers, and if editing is performed on the 14th layer, we map this to the corresponding layer in Qwen2.5-14B (which has 40 layers) using a proportional scaling method—i.e., $40 \times 14/28 = 20$th layer. Leveraging this layer-mapping approach, we conducted experiments on both Qwen2.5-7B and Qwen2.5-14B, with the results presented in Tab 8 and Tab 9. We found that as the backbone model is upgraded from Qwen2.5-7B to Qwen2.5-14B, the final performance of all methods is improved. Nevertheless, our method still achieves the most superior performance in most cases, which demonstrates that our approach remains applicable to larger-scale models. In addition, we observed that for Mu-Edit, the performance change (before and after editing) on Qwen2.5-14B is generally more significant than that on Qwen2.5-7B, especially on the SQUAD task. We hypothesize that this phenomenon is primarily attributed to the insufficient

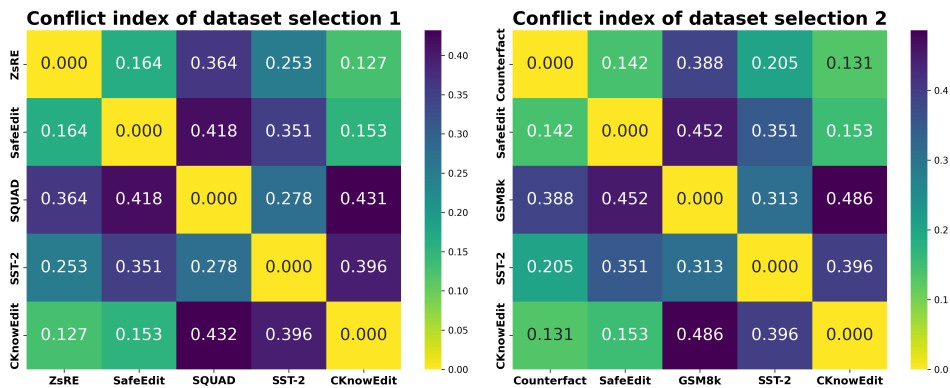

Figure 7: Conflict Index visualization

precision of the layer-mapping strategy. This observation also inspires our future research directions, which will focus on exploring interpretability and editing techniques tailored for larger-scale models.

## A.9 CASE STUDY

We selected several editing samples from the ZsRE, SQUAD and CKnowEdit datasets as case studies to analyze MEMIT, AlphaEdit and Mu-Edit's performance after multi-task editing. The results are displayed in Fig 5. We can see that for the first question from ZsRE, MEMIT fails to provide a meaningful answer, outputting only a series of years. Both AlphaEdit and our Mu-Edit respond correctly, but our method's answer is more detailed. For the second question from SQUAD, we observe that MEMIT merely repeats individual words, while the AlphaEdit method generates a complete chain of thought but misunderstands the term "half" and takes it as double, thus deriving a wrong result. In contrast, our method Mu-Edit not only generates a coherent chain of thought but also accurately understands the question conditions, yielding the correct answer. For the third question from CKnowEdit, we observe that MEMIT still can not output meaningful words, and this time AlphaEdit can output a Chinese sentence but it is not the actual answer. Our Mu-Edit can output true answer after multi-task editing.

## A.10 VISUALIZATION OF TASK CONFLICTS

In this section, we visualize the conflict index among five tasks. We adopt two dataset selections: the first is ZsRE, SafeEdit, SQUAD, SST-2 and CKnowEdit, the second is Counterfact, SafeEdit, GSM8k, SST-2 and CknowEdit. And we visualize both results in Fig 7. We can draw three conclusions from these two figures: (1) the conflict index of all the task pairs fall between 0.1 and 0.5, and more than 2/3 of the indicators exceed 0.2, which on the other hand justify that our low-rank decomposition of the matrix with 0.2 as the threshold is reasonable. (2) There is indeed a noticeable difference in conflict index between different tasks. For example, the conflict index between ZsRE, SafeEdit and CKnowEdit is less than 0.2, while the more difficult tasks such as SQUAD and GSM8k have a conflict index value higher than 0.35 between other tasks. The highest conflict index value belongs to GSM8k and CKnowEdit, reaching 0.486, which shows that SQUAD and GSM8k have higher conflicts with other tasks, so it is difficult to learn the corresponding knowledge while preserving the already-edited knowledge. And it also shows the rationality of finding the optimal editing order according to the conflict index between different tasks. (3) Under the conditions of two different dataset selections, we found that the heatmaps of the conflict index were highly similar. The conflict index of the same task's dataset is also very close to that of other datasets, which shows that our selected dataset is sufficient to represent the knowledge of a task.

## A.11 VERIFICATION OF OUR MU-EDIT ADDRESSES THE PROBLEM OF MULTIPLE SAMPLES PERFORMANCE DEGRADATION

We supplement experiments where Mu-Edit performs knowledge editing on datasets scaled from 500 samples to 1,000, 2,000, and 3,000 samples per task, and following the evaluation setting of

Table 10: Comparison results of additional ablation studies

| | ZsRE | | | | | | | | |
|---|---|---|---|---|---|---|---|---|---|
| Method | I-Specificity↑ | I-Generality↑ | I-Locality↑ | F-Specificity↑ | F-Generality↑ | F-Locality↑ | Spec change↑ | Gen change↑ | Loc change↑ |
| Mu-Edit | 0.9892 | 0.9375 | 0.6674 | 0.8845 | 0.8452 | 0.6321 | -0.1047 | -0.0923 | -0.0353 |
| w/o mean | 0.9826 | 0.9337 | 0.6641 | 0.8577 | 0.8185 | 0.6295 | -0.1249 | -0.1152 | -0.0346 |
| w/o variance | 0.9851 | 0.9344 | 0.6635 | 0.8650 | 0.8227 | 0.6260 | -0.1201 | -0.1117 | -0.0375 |

| | SafeEdit | | | SQUAD | | | SST-2 | | |
|---|---|---|---|---|---|---|---|---|---|
| Method | I-Harmful rate↑ | F-Harmful rate↑ | Harm change↑ | I-Acc↑ | F-Acc↑ | Acc change↑ | I-Acc↑ | F-Acc↑ | Acc change↑ |
| Mu-Edit | 0.4734 | 0.4043 | -0.0691 | 0.7867 | 0.7479 | -0.0388 | 0.9185 | 0.8838 | -0.0347 |
| w/o mean | 0.4708 | 0.3792 | -0.0916 | 0.7835 | 0.7255 | -0.0610 | 0.9172 | 0.8824 | -0.0348 |
| w/o variance | 0.4713 | 0.3836 | -0.0877 | 0.7829 | 0.7248 | -0.0581 | 0.9175 | 0.8828 | -0.0347 |

Table 11: Performance under different choice of $t$ and $\alpha$

| | $t$ parameter influence | | | | | | | | |
|---|---|---|---|---|---|---|---|---|---|
| $t$ value | ZsRE-Specificity | ZsRE-Generality | ZsRE-Locality | SafeEdit | SQUAD | SST-2 | CKnowEdit | Calculating Time | GPU Cost |
| 0.3 | 0.5911 | 0.5385 | 0.6433 | 0.3017 | 0.6852 | 0.7883 | 0.6981 | 124min | 214.5GB |
| 0.6 | 0.8447 | 0.7926 | 0.6391 | 0.3559 | 0.7226 | 0.8573 | 0.8064 | 160min | 291.1GB |
| 0.9 | 0.8845 | 0.8452 | 0.6321 | 0.4043 | 0.7479 | 0.8838 | 0.8505 | 238min | 330.6GB |
| 1.2 | 0.8221 | 0.7806 | 0.6182 | 0.3745 | 0.7158 | 0.8634 | 0.8297 | 287min | 387.8GB |
| 1.5 | 0.7234 | 0.6695 | 0.5996 | 0.3286 | 0.6660 | 0.8157 | 0.7668 | 356min | 443.5GB |

| | $\alpha$ parameter influence | | | | | | | | |
|---|---|---|---|---|---|---|---|---|---|
| $\alpha$ value | ZsRE-Specificity | ZsRE-Generality | ZsRE-Locality | SafeEdit | SQUAD | SST-2 | CKnowEdit | Calculating Time | GPU Cost |
| 0.5 | 0.7199 | 0.6881 | 0.5452 | 0.2909 | 0.6453 | 0.7822 | 0.6645 | 251min | 376.5GB |
| 0.75 | 0.8423 | 0.8004 | 0.6047 | 0.3715 | 0.7204 | 0.8538 | 0.8161 | 245min | 353.1GB |
| 1 | 0.8845 | 0.8452 | 0.6321 | 0.4043 | 0.7479 | 0.8838 | 0.8505 | 238min | 330.6GB |
| 1.5 | 0.8663 | 0.8215 | 0.6389 | 0.3895 | 0.7366 | 0.8684 | 0.8337 | 226min | 301.3GB |
| 2 | 0.8335 | 0.7669 | 0.6414 | 0.3592 | 0.6994 | 0.8401 | 0.7829 | 219min | 264.5GB |

D4S (Huang et al., 2024a), we only test the model performance until finishing editing on all samples. The results are shown in Tab 12. We observe that when the number of samples per task increases from 500 to 3,000, the editing performance only experiences a slight degradation. This demonstrates that our model maintains superior performance even under multi-sample editing scenarios.

## A.12 ADDITIONAL ANALYTICAL STUDIES

We conduct analytical experiments on the determination of $\mu$ values through mean and variance, as well as the selection of hyperparameters $\alpha$ and $t$ involved in the low-rank approximation. We first conducted separate analyses on the effects of removing the mean and replacing the mean of all tasks with 0.2 while retaining the variance, as well as removing the variance and retaining only the mean for the selection of $\mu$. The results show that both removing the mean and variance decline the final editing performance. This indicates that the conflict index considering both mean and variance of datasets from different tasks are helpful for determining the final threshold.

We further choose the number of $t$ as 0.3, 0.6, 0.9, 1.2 and 1.5 and $\alpha$ as 0.5, 0.75, 1, 1.5 and 2. We can find that when $t$ increases from 0.3 to 0.6, all seven metrics increase rapidly, but when $t$ reaches more than 0.6 to 0.9, we find that the increase become much more moderate, and as $t$ reaches more than 0.9 the performance begins to decline, the similar phenomenon arises with the increase of $\alpha$, reaching the best performance when $\alpha$ is 1. So finally we choose $t$ as 0.9 and $\alpha$ as 1.

Table 12: Editing Performance under increasing samples of data in one task

| | ZsRE | | | SafeEdit | SQUAD | SST-2 |
|---|---|---|---|---|---|---|
| Dataset Size | Specificity | Generality | Locality | Harmful rate | Acc | Acc |
| 500 | 0.9892 | 0.9375 | 0.6674 | 0.4734 | 0.7867 | 0.9185 |
| 1000 | 0.9826 | 0.9334 | 0.6641 | 0.4727 | 0.7825 | 0.9166 |
| 2000 | 0.9773 | 0.9296 | 0.6619 | 0.4713 | 0.7789 | 0.9163 |
| 3000 | 0.9722 | 0.9257 | 0.6602 | 0.4718 | 0.7735 | 0.9146 |

Table 13: Performance under the number of adjacent tasks during square norms minimization

| Adjacent tasks number | ZsRE-Spec | ZsRE-Gen | ZsRE-Loc | SafeEdit | SQUAD | SST-2 | CKnowEdit | Calculating Time | GPU Cost |
|---|---|---|---|---|---|---|---|---|---|
| 3 | 0.8845 | 0.8452 | 0.6321 | 0.4043 | 0.7479 | 0.8838 | 0.8505 | 238min | 330.6 GB |
| 4 | 0.8903 | 0.8443 | 0.6398 | 0.3996 | 0.7264 | 0.8826 | 0.8513 | 305min | 456.6 GB |
| 5 | 0.8714 | 0.8401 | 0.6452 | 0.3987 | 0.7126 | 0.8830 | 0.8466 | 360min | 585.4 GB |

### A.13 POSSIBLE LIMITATIONS OF MU-EDIT'S LOW-RANK APPROXIMATION METHOD

Overall, our Mu-Edit can greatly alleviate the conflict issues caused by multi-task editing. However, there are also possible limitations, mainly focusing on two aspects: 1. Is there a situation where the low-rank approximation method of Mu-Edit cannot completely resolve conflicts when the conflict index is too high? What is the overall performance in this situation? 2. The goal of Mu-Edit in the main text is to minimize the square norms of the $W$ matrix before and after updates for three consecutive adjacent tasks. Can the editing goal be set to four or more consecutive tasks?

For question 1, we need to calculate the rank reduction degree under the biggest conflict index pairs. We find that for the current experimental task setup, the conflict index between CknowEdit and GSM8k reached a maximum of 0.486. In this occasion if we want to reduce the conflict index below the threshold, we need to reduce the rank of 43.7%. According to our previous experiments, we were able to ensure that the single task editing effect did not significantly decrease even when the rank decreased by no more than 45%. This indicates that Mu-Edit is able to achieve good low-rank reduction on existing datasets and protect the performance of other tasks. But the only problem may be that we need to repeat the low-rank reduction process 6 times to successfully reduce the conflict index to the expected level, which increases the computational cost. When the conflict index is further increased, it may cause a greater computational burden. Our future research will focus on addressing this issue.

For question 2, we compare the performance of the task and the computational cost of minimizing three adjacent tasks, four adjacent tasks, and five adjacent tasks' square norms. We have found from Tab 13 that when we increase the number to 4, ZsRE-Specity, ZsRE-Locality and CKnowEdit shows a slight improvement, while there is a obvious performance decrease in SQUAD, and the computational cost also increases. And when we further increase the number to 5, we find that all performance stablizes or declines. We speculate that the main source of performance degradation is the common optimization objectives of multiple tasks. Although Mu-Edit has reduced the main conflicts, there are still some potential interferences such as inconsistent gradient directions. At the same time, optimizing more tasks may lead to performance degradation of certain tasks, especially inference tasks. Therefore, we ultimately set the number of consecutive tasks to 3.

### A.14 LLM USAGE AND ETHICS STATEMENT

LLMs were used to aid in the writing and polishing of the manuscript. Specifically, we used LLM to assist in refining the language, improving readability, and ensuring clarity in various sections of our paper. We must note that LLMs are not involved in the ideation, research methodology and experimental design.

In this study, no human subjects or animal experimentation was involved. All datasets were sourced in compliance with relevant usage guidelines, ensuring no validation of privacy. And due to the good security alignment of existing large language models, the knowledge edited by SafeEdit may cause the model to relearn unsafe responses, but we only use it for scientific research.

