# OpenReview forum: "MuEdit: A Lightweight yet Effective Multi-task Model Editing Method"
_ICLR.cc/2026/Conference — Submitted to ICLR 2026_

### Official Review · Reviewer_XQXJ · 2025-10-16

**Soundness:** 3
**Presentation:** 3
**Contribution:** 2
**Rating:** 4
**Confidence:** 4

**Summary:**

This paper points out that existing knowledge editing methods cannot effectively handle multi-task editing, and conflicts exist between different tasks, which affects editing performance. This paper proposes a multi-task editing framework that uses two complementary strategies to resolve multi-task conflicts.

**Strengths:**

1. This paper is well-written, with clear logic and concise readability.

2. It conducts extensive experiments, including comparative experiments on models of different types and scales.

3. The authors provide a wide range of comprehensive evaluation metrics.

**Weaknesses:**

1. First, the paper points out that Mu-Edit relies on low-rank decomposition to achieve editing across multiple tasks, but lacks comparative results with full-model fine-tuning and LoRA.

2. Second, the motivation is insufficient. Knowledge editing is often used for low-cost knowledge updates. It remains unclear whether the setting of performing knowledge updates across multiple tasks is reasonable, and what advantages this setting offers over full-model fine-tuning or LoRA.

3. Finally, previous works (such as D4S [1] and AlphaEdit [2]) have addressed the issue of model performance degradation after editing multiple samples. Based on the existing experimental results, Mu-Edit fails to demonstrate this capability, which casts doubts on its practical application.

**References**

[1] Reasons and Solutions for the Decline in Model Performance after Editing

[2] Alphaedit: Null-space constrained knowledge editing for language models

**Questions:**

See Weaknesses.

---

> ### Author Response · Authors · 2025-11-19
> **Reply to Reviewer XQXJ: Part1**
>
> Thanks for raising a series of valuable questions about the paper. Below, I will address these questions one by one:
>
> W1&W2:
>
> To further validate our method, we conduct additional comparative experiments with Fine-tuning and LoRA under the Best Order and Default Order settings, whose results are summarized in the table below.  As you can see that the Fine-tuning method still incurs significant performance degradation before and after editing. LoRA reduces such performance loss through sparse parameter adaptation, yet its performance in Immediate testing is generally inferior to that of Fine-tuning. **In contrast, our method achieves improvements in pre-editing performance, post-editing performance, and the performance decay between pre- and post-editing on most tasks compared to both LORA and Fine-tuning**. This demonstrates that our approach is more effective than the aforementioned two methods for multi-task knowledge updating. Regarding the rationality of knowledge updating across multiple tasks as raised in your comment, in practical scenarios, we often need to maintain or update various functionalities of a model. For instance, a conversational assistant may require updates to both its dialogue understanding capabilities and safety performance. In such cases, utilizing Mu-Edit for knowledge updating is clearly reasonable.
>
> | Backbone | Method | I-Spec | I-Gen | I-Loc | F-Spec | F-Gen | F-Loc | Spec change | Gen change | Loc change |
> |:--------:|:------:|:------:|:-----:|:-----:|:------:|:-----:|:-----:|:-----------:|:----------:|:----------:|
> | | Fine-tuning(DO) | 0.8454         | 0.6541         | 0.4417      | 0.5750        | 0.4223       | 0.3136     | -0.2704      | -0.2318    | -0.1281    |
> | | LORA(DO)   | 0.7789         | 0.6331         | 0.4269      | 0.5832         | 0.4642       | 0.3143     | -0.1957     | -0.1689    | -0.1126    |
> | | Mu-Edit(DO)   | 0.9883         | 0.9366         | 0.6669      | 0.8197       | 0.7995       | 0.5684     | -0.1686     | -0.1371    | -0.0985    |
> | Llama3-8B| Fine-tuning(BO) | 0.8503         | 0.6596         | 0.4484      | 0.6462        | 0.5397       | 0.3819     | -0.2041     | -0.1199    | -0.0665    |
> | | LoRA(BO)   | 0.8131         | 0.6794         | 0.4875      | 0.6668        | 0.5677       | 0.4064     | -0.1463     | -0.1217    | -0.0811    |
> | | Mu-Edit(BO)   | **0.9892**         | **0.9375**         | **0.6674**      | **0.8845**        | **0.8452**       | **0.6321**     | **-0.1047**     | **-0.0923**    | **-0.0353**    |
>
> | Backbone | Method | I-Harm| F-Harm | Harm change | I-Acc | F-Acc | Acc change | I-Acc | F-Acc | Acc change |
> |:--------:|:------:|:--------------:|:--------------:|:-----------:|:-----:|:-----:|:----------:|:-----:|:-----:|:----------:|
> | | Fine-tuning(DO) | 0.5684         | 0.3013         | -0.2671     | 0.7477        | 0.6285       | -0.1192    | 0.8696      | 0.8378     | **-0.0318**    |
> | | LORA(DO)   | 0.4467         | 0.2869         | -0.1598     | 0.7310        | 0.6105       | -0.1205     | 0.8897      | 0.8553     | -0.0344    |
> | | Mu-Edit(DO)   | 0.4706        | 0.3648         | -0.1058     | 0.7867        | 0.7252       | -0.0615    | 0.9185      | 0.8694     | -0.0491    |
> | Llama3-8B| Fine-tuning(BO) | **0.5725**         | 0.3521         | -0.2204     | **0.7992**        | 0.6583       | -0.1409    | 0.9163      | 0.8782     | -0.0381    |
> | | LORA(BO)   | 0.5399         | 0.4012         | -0.1387      | 0.7672        | 0.6998       | -0.0674    | 0.9142       | 0.8801     | -0.0341    |
> | | Mu-Edit(BO)   | 0.4734         | **0.4043**         | **-0.0691**     | 0.7867        | **0.7479**       | **-0.0388**    | **0.9185**      | **0.8838**     | -0.0347    |

---

> > ### Author Response · Authors · 2025-11-19
> > **Reply to Reviewer XQXJ: Part2**
> >
> > W3: Maintaining performance after editing multiple samples:
> >
> > We supplement experiments where Mu-Edit performs knowledge editing on datasets scaled from 500 samples to 1,000, 2,000, and 3,000 samples per task, with the results presented in the table below. **We observe that when the number of samples per task increases from 500 to 3,000, the editing performance only experiences a slight degradation**. This demonstrates that our model maintains superior performance even under multi-sample editing scenarios.
> >
> > |              |             |      ZsRE          |          |   SafeEdit   |  SQUAD |  SST-2 |
> > |--------------|:-----------:|:----------:|:--------:|:------------:|:------:|:------:|
> > | Dataset Size | Specificity | Generality | Locality | Harmful rate |   Acc  |   Acc  |
> > | 500          |    0.9892   |   0.9375   |  0.6674  |    0.4734    | 0.7867 | 0.9185 |
> > | 1000         |    0.9826   |   0.9334   |  0.6641  |    0.4727    | 0.7825 | 0.9166 |
> > | 2000         |    0.9773   |   0.9296   |  0.6619  |    0.4713    | 0.7789 | 0.9163 |
> > | 3000         |    0.9722   |   0.9257   |  0.6602  |    0.4718    | 0.7735 | 0.9146 |
> >
> > All of the above is the comprehensive response to all your inquiries. We sincerely appreciate your attention to our work. Should you have any further questions or require clarification on any points that may not have been fully addressed, please do not hesitate to inform us.

---

> > > ### Comment · Reviewer_XQXJ · 2025-11-20
> > >
> > > Thank the authors for their response. I still have some questions.
> > >
> > > The authors have demonstrated that the performance remains excellent after editing multiple samples by increasing the number of samples. However, since the authors have not released the complete code, I would like to confirm whether your evaluation method involves evaluating immediately after editing one sample (following methods such as ROME and MEMIT) or evaluating after editing all samples (following the D4S method). These two different evaluation methods lead to significant differences in the final results and also have a substantial impact on the reliability of the editing method.

---

> > > > ### Author Response · Authors · 2025-11-20
> > > > **Reply to your question of evaluation:**
> > > >
> > > > Thanks for your question, in fact, we conduct our experiment following the setting of D4S method. Specifically, we evaluate our model performance after editing all the 500, 1000, 2000 and 3000 pieces of data in the four experiments.

---

> > > > ### Author Response · Authors · 2025-11-27
> > > > **Confirming whether our clarification has addressed your concerns**
> > > >
> > > > We add the comparison results of performance degradation after editing multiple samples with AlphaEdit and MEMIT, the latter set as the baseline of not handling multiple samples performance degradation, and the results are illustrated in the three tables below:
> > > >
> > > > | Method | Dataset Size | Specificity | ZsRE Generality | ZsRE Locality | SafeEdit Harmful rate | SQUAD Acc | SST-2 Acc |
> > > > |--------|--------------|-------------|-----------------|---------------|-----------------------|-----------|-----------|
> > > > | | 500          | 0.7009      | 0.6817          | 0.4962        | 0.3781                | 0.5712    | 0.7335    |
> > > > | MEMIT    | 1000         | 0.6571      | 0.6294          | 0.4337        | 0.2885                | 0.5413    | 0.7004    |
> > > > | | 2000         | 0.4793      | 0.3246          | 0.3384       | 0.1901                | 0.3776    | 0.6057    |
> > > > | | 3000         | 0.2414      | 0.1297          | 0.2229        | 0.1103                | 0.2618    | 0.5513    |
> > > >
> > > > | Method | Dataset Size | Specificity | ZsRE Generality | ZsRE Locality | SafeEdit Harmful rate | SQUAD Acc | SST-2 Acc |
> > > > |--------|--------------|-------------|-----------------|---------------|-----------------------|-----------|-----------|
> > > > | | 500          | 0.9889      | 0.9377          | 0.6628        | 0.4664                | 0.7883    | 0.9167    |
> > > > | AlphaEdit    | 1000         | 0.9855      | 0.9338          | 0.6606        | 0.4648                | 0.7842    | 0.9153    |
> > > > | | 2000         | 0.9822      | 0.9315          | 0.6582        | 0.4619                | 0.7818    | 0.9148    |
> > > > | | 3000         | 0.9780      | 0.9283          | 0.6570        | 0.4611                | 0.7797    | 0.9129    |
> > > >
> > > > | Method | Dataset Size | Specificity | ZsRE Generality | ZsRE Locality | SafeEdit Harmful rate | SQUAD Acc | SST-2 Acc |
> > > > |--------|--------------|-------------|-----------------|---------------|-----------------------|-----------|-----------|
> > > > | | 500          | 0.9892      | 0.9375          | 0.6674        | 0.4734                | 0.7867    | 0.9185    |
> > > > | Mu-Edit    | 1000         | 0.9826      | 0.9334          | 0.6641        | 0.4727                | 0.7825    | 0.9166    |
> > > > | | 2000         | 0.9773      | 0.9296          | 0.6619        | 0.4713                | 0.7789    | 0.9163    |
> > > > | | 3000         | 0.9722      | 0.9257          | 0.6602        | 0.4718                | 0.7758    | 0.9146    |
> > > >
> > > > Our experimental results demonstrate that both AlphaEdit and Mu-Edit can mitigate the performance degradation that occurs after editing multiple examples. Specifically, AlphaEdit may exhibit a slightly better effect in alleviating such degradation following repeated edits. Nevertheless, compared with traditional methods like MEMIT, **our approach still achieves a substantial mitigation of performance degradation after multiple edits and is capable of handling 3,000 single-domain multiple editing tasks**.
> > > >
> > > > Should you have any other concerns, please feel free to raise them at your earliest convenience, and we will provide a prompt response. If you are satisfied with our responses, we would be highly grateful if you could consider increasing the score, thank you!

---

> > > > > ### Author Response · Authors · 2025-11-27
> > > > > **We view revise our PDF further to add additional references:**
> > > > >
> > > > > We have identified a lack of additional references related to basic knowledge editing and lifelong editing in our manuscript. To ensure academic rigor and compliance with scholarly standards, we will supplement these references in the revised PDF submission prior to the deadline, including the following works:
> > > > >
> > > > > **[1] Commonsense knowledge editing based on free-text in llms**
> > > > >
> > > > > **[2] Perturbation-restrained sequential model editing**
> > > > >
> > > > > **[3] If an LLM Were a Character, Would It Know Its Own Story? Evaluating Lifelong Learning in LLMs**
> > > > >
> > > > > **[4] Model Editing Harms General Abilities of Large Language Models: Regularization to the Rescue**
> > > > >
> > > > > These papers conduct in-depth analyses from the perspectives of lifelong knowledge editing, common sense knowledge editing, and the evaluation of the impact of existing editing methods on models’ overall performance. **They represent excellent contributions to the field, and we have drawn inspiration from certain aspects of these works—such as SVD regularization and layer-wise localization analysis—for specific components of our study**.
> > > > >
> > > > > Finally, we welcome any suggestions you may have to further enhance the quality and rigor of our paper. We will be happy to address them promptly and revise the paper accordingly.

---

> ### Author Response · Authors · 2025-11-23
> **We have revised our PDF according to your advice.**
>
> We have provided additional analysis based on your questions and revised our paper. Specifically, we add the reference of D4S method in our additional related work in **A.1** in our revised PDF. And the settings of additional experiments and results to show that our method can solve the performance degradation of multi sample knowledge editing are displayed in **A.11** in our revised PDF. Please kindly check it.
>
> We sincerely appreciate your willingness to share whether our revisions have adequately addressed the concerns you raised during the review process. Should our responses meet your expectations and you feel satisfied with the improvements made, we would be extremely grateful if you could kindly consider increasing your evaluation score accordingly, thanks!

---

### Official Review · Reviewer_7WTo · 2025-10-26

**Soundness:** 3
**Presentation:** 2
**Contribution:** 3
**Rating:** 6
**Confidence:** 3

**Summary:**

This paper proposes MuEdit, a lightweight and effective method for multi-task model editing. The authors argue that existing model editing approaches suffer from strong interference when updating multiple tasks simultaneously. To address this, they introduce a novel metric called the Conflict Index to quantify conflicts between task-specific editing objectives.
Based on this metric, they design two strategies Optimal Editing Order Selection, and Conflict-Guided Low-Rank Matrix Approximation to solve this problem.
Extensive experiments on multiple benchmarks and two model (Llama3-8B and GPT2-XL) demonstrate that MuEdit outperforms state-of-the-art methods such as ROME, MEMIT, AlphaEdit, and AnyEdit, while maintaining strong general-domain capabilities.

**Strengths:**

1.	Novel problem formulation – The paper is the first to explicitly define and analyze multi-task model editing from a null-space conflict perspective.
2.	The theoretical foundation based on linear algebra (null-space and rank analysis) is sound and logically consistent, which is an interpretable approach.
3.	This paper Covers five heterogeneous tasks, Includeing ablation studies, sensitivity analysis, and significance testing (p < 0.05). MuEdit achieves substantial improvements in multi-task editing and maintains general-domain abilities better than all baselines.

**Weaknesses:**

1.	Although the Conflict Index is an interesting idea, it is heuristic and lacks a rigorous theoretical connection to optimization conflicts (e.g., gradient interference or Fisher information).
2.	The “optimal editing order” involves a factorial search over tasks (O(N!)); the paper does not clarify how this is handled in practice.
3.	The method assumes all tasks are known beforehand; it is unclear how Mu-Edit performs when new tasks arrive incrementally.
4.	Results are shown on GPT2-XL and Llama3-8B; it remains uncertain whether the conclusions hold for larger models like Llama3-70B.

**Questions:**

1. How scalable is the Conflict Index computation and order search when the number of tasks exceeds 10?

2. Does low-rank approximation reduce the model’s knowledge capacity, potentially leading to long-term forgetting?

---

> ### Author Response · Authors · 2025-11-19
> **To Reviewer 7WTo: Part1**
>
> Thanks for raising a series of valuable questions about the paper. Below, I will address these questions one by one:
>
> W1:
> Your comment is highly valuable. In fact, we currently face challenges in directly establishing a theoretical relationship between the proposed conflict index and gradient interference or Fisher information. We will strive to address this limitation in our future research. Actually, many outstanding works in the field have first proposed concepts and later supplemented them with comprehensive theoretical proofs. On the other hand, we can empirically illustrate the inherent correlation between the conflict index proposed in this paper and gradients from an experimental perspective:
>
> The conflict index is designed based on fundamental insight: **if the update gradient directions of two tasks are inconsistent, updating one task will interfere with the other**. In model editing, our goal is to enable the model to memorize new knowledge $K_e$ through parameter updates $\Delta\theta\$. If the dot product of the optimal update $\Delta\theta_A$ for Task A and $\Delta\theta_B$ for Task B is negative (i.e., their directions are opposite), it will be extremely difficult to satisfy both tasks simultaneously. If a vector $\Delta\theta$ lies in the null space of the knowledge matrix $K_A$ of Task A, then updating parameters along the direction of $\Delta\theta$ will not alter the model's output corresponding to Task A. Therefore, if we can constrain the update for Task B within the null space of Task A, editing Task B will not undermine the already learned knowledge of Task A. The conflict index $C(K_i, K_j)$ defined in this paper measures the size of the intersection of the null spaces of two tasks relative to their individual null spaces. This intersection is precisely the space composed of update directions that do not interfere with either task.
>
> **- Small conflict index → Large common null space** → Numerous update directions exist to effectively edit one task without interfering with the other at all.
>
> **- Large conflict index → Small common null space** → Almost no update direction can edit a new task without disrupting existing tasks. Any effective edit will inevitably lead to "catastrophic forgetting."
>
> W2 and W3: To address the issues that Best Order cannot be directly applied when the number of tasks is excessively large in practice and fails to dynamically update in a timely manner upon the arrival of new tasks, **we propose a Greedy Search Order method in our practical research**. On one hand, regarding the original optimal editing order with a time complexity of O(N!), we note that when N < 9, the number of computations will not exceed 40,320, and Best Order can still be used within a reasonable computational cost. When N >= 9, we adopt the Greedy Search Order algorithm with a time complexity of O(N^2) in practice. On the other hand, the Greedy Search Order is more suitable for scenarios where new tasks arrive incrementally: Specifically, when a new task arrives, the greedy algorithm only needs to construct a $K$ matrix using partial representative data of the new task, then compare the Conflict Index between the $K$ matrix of the new task and all original tasks one by one. The new task is inserted at the position where the sum of conflict indices with the preceding and subsequent tasks is minimized, without the need to recalculate the Conflict Index relationships between the original tasks. This makes it more adaptable to scenarios where knowledge is continuously added in practical applications.

---

> ### Author Response · Authors · 2025-11-19
> **To Reviewer 7WTo: Part2**
>
> W4:
> To clarify, research in model editing, including the field of interpretability, **primarily achieves intended model improvements through neuron-level studies and control on small-scale models**. This is grounded in the profound advancements made in the interpretability of small models (typically 8B parameters or smaller) in recent years. Notably, prominent model editing works such as AlphaEdit and AnyEdit have predominantly focused on models like Llama3-8B and Qwen2.5-7B. On the other hand, as demonstrated in Table 2 of the main text, computing the $K$-matrix on the Llama3-8B model using 100,000 data samples already requires 330.6 GB of memory—making the computational overhead for larger models (e.g., 70B parameters) currently prohibitive.
>
> Nevertheless, we infer that your question may pertain to **whether a scaling law similar to that of fine-tuning (where overall performance improves with larger model sizes) exists in model editing, and whether our Mu-Edit method can still outperform other approaches on larger models**. So we conduct experiments under the same setting on Qwen2.5-7B and Qwen2.5-14B. Given the limited existing research on identifying the most effective layers for model editing 14B-scale models, we propose a layer-mapping strategy: for instance, Qwen2.5-7B comprises 28 layers, and if editing is performed on the 14th layer, we map this to the corresponding layer in Qwen2.5-14B (which has 40 layers) using a proportional scaling method—i.e., 40 × 14/28 = 20th layer. Leveraging this layer-mapping approach, we conducted experiments on both Qwen2.5-7B and Qwen2.5-14B, with the results presented below:
>
> | Backbone | Method | I-Spec | I-Gen | I-Loc | F-Spec | F-Gen | F-Loc | Spec change | Gen change | Loc change |
> |:--------:|:------:|:------:|:-----:|:-----:|:------:|:-----:|:-----:|:-----------:|:----------:|:----------:|
> | | AlphaEdit(DO) | 0.9900 | 0.9363 | 0.6617 | 0.3031 | 0.2366 | 0.3615 | -0.6869 | -0.6997 | -0.3002 |
> | | AnyEdit(DO) | 0.9904 | 0.9382 | 0.6698 | 0.4127 | 0.3298 | 0.4219 | -0.5777 | -0.6084 | -0.2479 |
> | | Mu-Edit(DO) | 0.9895 | 0.9377 | 0.6701 | 0.8199 | 0.8006 | 0.5703 | -0.1696 | -0.1371 | -0.0998 |
> | Qwen2.5-7B | AlphaEdit(BO) | 0.9904 | 0.9381 | 0.6637 | 0.5776 | 0.5145 | 0.4482 | -0.4128 | -0.4236 | -0.2155 |
> | | AnyEdit(BO) | **0.9919** | **0.9408** | 0.6670 | 0.7142 | 0.6807 | 0.5369 | -0.2777 | -0.2601 | -0.1301 |
> | | Mu-Edit(BO) | 0.9911 | 0.9404 | **0.6706** | **0.8862** | **0.8458** | **0.6337** | **-0.1049** | **-0.0946** | **-0.0369** |
>
> | Backbone | Method | I-Harm| F-Harm | Harm change | I-Acc | F-Acc | Acc change | I-Acc | F-Acc | Acc change |
> |:--------:|:------:|:--------------:|:--------------:|:-----------:|:-----:|:-----:|:----------:|:-----:|:-----:|:----------:|
> | | AlphaEdit(DO) | 0.4384 | 0.2801 | -0.1583 | 0.7984 | 0.7119 | -0.0865 | 0.9209 | 0.8702 | -0.0507 |
> | | AnyEdit(DO) | 0.4615 | 0.3022 | -0.1593 | 0.8032 | 0.7264 | -0.0768 | 0.9236 | 0.8725 | -0.0511 |
> | | Mu-Edit(DO) | 0.4725 | 0.3668 | -0.1057 | 0.8095 | 0.7539 | -0.0556 | 0.9261 | 0.8832 | -0.0429 |
> | Qwen2.5-7B | AlphaEdit(BO) | 0.4709 | 0.3676 | -0.1033 | 0.7997 | 0.7203 | -0.0788 | 0.9278 | 0.8822 | -0.0456 |
> | | AnyEdit(BO) | 0.4732 | 0.3878 | -0.0854 | 0.8065 | 0.7444 | -0.0621 | 0.9302 | 0.8814 | -0.0488 |
> | | Mu-Edit(BO) | **0.4769** | **0.4058** | **-0.0711** | **0.8191** | **0.7782** | **-0.0409** | **0.9337** | **0.8974** | **-0.0363** |
>
>
> The results of Qwen2-14B are illustrated in Part 3 due to the limit of word counts.

---

> ### Author Response · Authors · 2025-11-19
> **To Reviewer 7WTo: Part3**
>
> | Backbone | Method | I-Spec | I-Gen | I-Loc | F-Spec | F-Gen | F-Loc | Spec change | Gen change | Loc change |
> |:--------:|:------:|:------:|:-----:|:-----:|:------:|:-----:|:-----:|:-----------:|:----------:|:----------:|
> | | AlphaEdit(DO) | 0.9914         | 0.9385         | 0.6689      | 0.3144        | 0.2453       | 0.3661     | -0.6770| -0.6832    | -0.3028    |
> | | AnyEdit(DO)   | 0.9923         | 0.9406         | 0.6702      | 0.4230         | 0.3374       | 0.4275     | -0.5693     | -0.6032    | -0.2427    |
> | | Mu-Edit(DO)   | 0.9914         | 0.9413         | 0.6713      | 0.8218        | 0.8041       | 0.5748     | -0.1696     | -0.1372    | -0.0965    |
> | Qwen2.5-14B| AlphaEdit(BO) | 0.9921         | 0.9409         | 0.6685      | 0.5802        | 0.5195       | 0.4503     | -0.4119     | -0.4214    | -0.2182    |
> | | AnyEdit(BO)   | **0.9936**         | 0.9421         | 0.6704      | 0.7189        | 0.6855       | 0.5398     | -0.2747     | -0.2566    | -0.1306    |
> | | Mu-Edit(BO)   | 0.9932         | **0.9425**         | **0.6721**      | **0.8898**        | **0.8493**       | **0.6366**     | **-0.1034**     | **-0.0932**    | **-0.0355**    |
>
> | Backbone | Method | I-Harm| F-Harm | Harm change | I-Acc | F-Acc | Acc change | I-Acc | F-Acc | Acc change |
> |:--------:|:------:|:--------------:|:--------------:|:-----------:|:-----:|:-----:|:----------:|:-----:|:-----:|:----------:|
> | | AlphaEdit(DO) | 0.4563         | 0.3115         | -0.1448     | 0.8102        | 0.7256       | -0.0846    | 0.9295      | 0.8813     | -0.0482    |
> | | AnyEdit(DO)   | 0.4704         | 0.3448         | -0.1256     | 0.8231        | 0.7481       | -0.0750     | 0.9352      | 0.8847     | -0.0505    |
> | | Mu-Edit(DO)   | 0.4818         | 0.3702         | -0.1122     | 0.8333        | 0.7688       | -0.0645    | 0.9422      | 0.8903     | -0.0519    |
> | Qwen2.5-14B| AlphaEdit(BO) | 0.4809         | 0.3698         | -0.1111     | 0.8164        | 0.7385       | -0.0779    | 0.9436      | 0.8925     | -0.0511    |
> | | AnyEdit(BO)   | 0.4845         | 0.3987         | -0.0858      | 0.8483        | 0.7702       | -0.0781    | 0.9490       | 0.9003     | -0.0487    |
> | | Mu-Edit(BO)   | **0.4892**         | **0.4174**         | **-0.0718**     | **0.8575**        | **0.8084**       | **-0.0491**    | **0.9511**      | **0.9122**     | **-0.0389**    |
>
> We found that as the backbone model is upgraded from Qwen2.5-7B to Qwen2.5-14B, the final performance of all methods is improved. **Nevertheless, our method still achieves the most superior performance in most cases, which demonstrates that our approach remains applicable to larger-scale models**. In addition, we observed that for Mu-Edit, the performance change (before and after editing) on Qwen2.5-14B is generally more significant than that on Qwen2.5-7B, especially on the SQUAD task. We hypothesize that this phenomenon is primarily attributed to the insufficient precision of the layer-mapping strategy. This observation also inspires our future research directions, which will focus on exploring interpretability and editing techniques tailored for larger-scale models.
>
> Q1:
> In fact, the majority of our computation time is spent on $K$-matrix calculation: for a dataset of 100,000 samples, the computation takes approximately 150–170 minutes. In contrast, the SVD decomposition of a $K$-matrix, utilizing the randomized SVD algorithm with distributed optimization, only requires about 1–2 minutes. The calculation of the Conflict Index primarily involves identifying the common null space of the SVD-decomposed individual $K$-matrices and merged $K$-matrices, with its time overhead largely concentrated on the SVD decomposition step. Although the complexity of computing the Conflict Index is proportional to the square of the number of tasks (N), **when N is slightly greater than 10, the total computation time is still around 200 minutes (resulting in an overall time of approximately 300–400 minutes)**. The memory overhead is nearly identical to that of calculating the Conflict Index for 5 tasks. Significant increases in time cost only occur when N > 20.
>
> For computing the optimal editing order, when N = 5, the brute-force search method to find the optimal order takes about 20–30 seconds. Given that the time complexity of brute-force search is O(N!), the computation time for N = 10 would exceed 600,000 seconds (≈167 hours), which is clearly infeasible. Therefore, when the number of tasks exceeds 10, the greedy search order should be adopted instead of the brute-force-based best order. In this case, the time overhead is approximately 200 seconds, which is fully acceptable.
>
> In summary, **the Conflict Index calculation is fully scalable when N is greater than 10 but less than 20**. Additionally, **the order search process is also scalable when the greedy search order replaces the best order (brute-force search) for N > 10**.

---

> > ### Author Response · Authors · 2025-11-19
> > **To Reviewer 7WTo: Part4**
> >
> > Q2:
> > Regarding the question of "knowledge capacity" you raised, we understand it may refer to the preservation of general capabilities or the performance of individual tasks. In terms of preserving general capabilities, Figure 3 in the main paper demonstrates that compared to all previous methods, **our Mu-Edit only experiences a slight degradation in performance on general domains after editing 5 tasks**. This indicates that the capacity for retaining general capabilities has not diminished. For the performance on individual tasks, Figure 4 in the main paper shows that the model's performance on individual tasks remains nearly stable until the rank reduction reaches 45%. Furthermore, in Appendix A.7, we illustrate that for the most conflicting task pair, **the rank reduction ratio is 43.7%, which does not exceed 45%**. Collectively, the above results confirm that the **low-rank reduction method barely impairs the overall knowledge capacity of the model**.
> >
> > All of the above is the comprehensive response to all your inquiries. We sincerely appreciate your attention to our work. Should you have any further questions or require clarification on any points that may not have been fully addressed, please do not hesitate to inform us.

---

> > > ### Author Response · Authors · 2025-11-26
> > > **To Reviewer 7WTo: Part5**
> > >
> > > Supplementary Note for W4:
> > >
> > > Building upon the Qwen2.5-14B backbone, we further extended our experiments to the **Qwen2.5-32B** model. The selection of edit layers adopted a layer mapping strategy similar to that used for Qwen2.5-32B. While the 32B model does not reach the 70B scale you mentioned, **it is sufficiently large to verify whether Mu-Edit is generalizable to relatively large-scale models**. During the actual experiments, we observed that estimating $K$ with 100,000 data samples exceeded the GPU memory budget when the backbone was Qwen2.5-32B. Consequently, we opted to estimate $K$ using 50,000 data samples and conducted comparative experiments between Mu-Edit and all other baseline methods. The results are presented in the following table.
> > >
> > > | Backbone | Method       | I-Spec | I-Gen | I-Loc | F-Spec | F-Gen | F-Loc | Spec change | Gen change | Loc change |
> > > |:--------:|:------------:|:------:|:-----:|:-----:|:------:|:-----:|:-----:|:-----------:|:----------:|:----------:|
> > > |          | AlphaEdit(DO) | 0.9917 | 0.9389 | 0.6689 | 0.3181 | 0.2448 | 0.3672 | -0.6736     | -0.6941    | -0.3017    |
> > > |          | AnyEdit(DO)   | 0.9923 | 0.9411 | 0.6721 | 0.4298 | 0.3712 | 0.4464 | -0.5625     | -0.5699    | -0.2257    |
> > > |          | Mu-Edit(DO)   | 0.9909 | 0.9411 | 0.6718 | 0.7986 | 0.7913 | 0.5814 | -0.1923     | -0.1498    | -0.0904    |
> > > | Qwen2.5-32B | AlphaEdit(BO) | 0.9921 | 0.9428 | 0.6755 | 0.6103 | 0.5286 | 0.4778 | -0.3818     | -0.4650    | -0.4650    |
> > > |          | AnyEdit(BO)   | **0.9941** | **0.9432** | **0.6787** | 0.7698 | 0.7801 | 0.5662 | -0.2243     | -0.3770    | -0.1125    |
> > > |          | Mu-Edit(BO)   | 0.9925 | 0.9417 | 0.6759 | **0.8863** | **0.8482** | **0.6357** | **-0.1062**     | **-0.0935**    | **-0.0402**    |
> > >
> > > | Backbone | Method       | I-Harm | F-Harm | Harm change | I-Acc | F-Acc | Acc change | I-Acc | F-Acc | Acc change |
> > > |:--------:|:------:|:--------------:|:--------------:|:-----------:|:-----:|:-----:|:----------:|:-----:|:-----:|:----------:|
> > > | | AlphaEdit(DO) | 0.4676 | 0.3521 | -0.1155     | 0.8332      | 0.7514      | -0.0818          | 0.9317      | 0.8924      | -0.0393          |
> > > | | AnyEdit(DO)   | 0.4815 | 0.3871 | -0.0944     | 0.8411      | 0.7708      | -0.0703          | 0.9404      | 0.8996      | -0.0408          |
> > > | | Mu-Edit(DO)   | 0.4832 | 0.3993 | -0.0839     | 0.8482      | 0.7732      | -0.0750          | 0.9450      | 0.9001      | -0.0449          |
> > > | Qwen2.5-32B | AlphaEdit(BO) | 0.4872 | 0.4075 | -0.0797     | 0.8395      | 0.7657      | -0.0738          | 0.9492      | 0.9143      | -0.0349          |
> > > | | AnyEdit(BO)   | **0.4907** | 0.4098 | -0.0809     | 0.8558      | 0.7912      | -0.0646          | **0.9523**      | **0.9168**      | -0.0355          |
> > > | | Mu-Edit(BO)   | 0.4883 | **0.4156** | **-0.0727**     | **0.8612**      | **0.8096**      | **-0.0546**          | 0.9481      | 0.9154      | **-0.0327**          |
> > >
> > > We observe that when Qwen2.5-32B is used as the backbone model, the prominent advantages of our Mu-Edit over AlphaEdit and AnyEdit are somewhat diminished, and suboptimal Immediate (I-) test or Final (F-) test results are observed on many tasks. Furthermore, a performance degradation phenomenon is observed on certain tasks (e.g., ZsRE) when Qwen2.5-32B is used as the backbone compared to the 14B counterpart. We hypothesize that this may stem from the reduced amount of data used for estimating $K$ or potential limitations of the layer mapping method. However, overall, our model still **achieves the smallest performance drop before and after editing across all tasks**, demonstrating that Mu-Edit can still yield favorable results even at the 32B scale.
> > >
> > > Through all the aforementioned experiments, we hope them can address your concerns. If any doubts still remain, please do not hesitate to point them out, and we will provide further clarifications promptly. If you are satisfied with our responses, we would be highly grateful if you could consider increasing the score or enhancing the confidence level. Thank you!

---

> ### Author Response · Authors · 2025-11-23
> **We have revised our PDF according to your advice.**
>
> We have provided additional analysis based on your questions and revised our paper. Specifically, the Greedy Search method and algorithm to tackle unpredictable knowledge updates and unaffordable computational complexity, as well as the time cost of conflict index calculation and order search is added in **Part 4.5** in our PDF in the main part of our paper.  The results of our Mu-Edit can generalize to larger-scale models are added in **A.8** in our PDF in the appendix. Please kindly check it. We sincerely appreciate your willingness to share whether our revisions have adequately addressed the concerns you raised during the review process.

---

### Official Review · Reviewer_4N7w · 2025-10-29

**Soundness:** 2
**Presentation:** 2
**Contribution:** 2
**Rating:** 4
**Confidence:** 4

**Summary:**

The author proposed a novel concept termed the Conflict Index, which quantifies the degree of conflict between the editing objectives of two tasks. Building on this idea, the author introduced a method that integrates two key strategies: 1) optimal edit path identification; 2) a low-rank matrix approximation method based on the conflict index to expand the null-space dimension.

**Strengths:**

- The author provides a clear formulation of the multi-task editing problem and introduces the Conflict Index, an insightful and valuable concept.
- The idea of leveraging the common null space and employing low-rank matrix decomposition to mitigate task conflicts is both inspiring and technically interesting.

**Weaknesses:**

- The paper strongly lacks analysis and experiment to support its idea.

(1) No further experiment to support the key observation of this paper, which is that during sequential multitask editing, the new knowledge matrix Kn compresses the null space of Kn−1 (in Sec. 3.2) after the teaser figure.

(2) In the Sec. 4.1 and the appendix, the main experiment was still conducted on the Llama3-8B and GPT2-XL, which is a pretty old combination. The author should add more experiments on SOTA LLMs like Qwen2.5.

- The proposed method mainly addresses the sequential editing scenario, which corresponds to lifelong model editing in practice. However, the Best Order concept introduced in Sec. 3.3.1 is not realistic in real-world applications, as the future knowledge to be edited is inherently unpredictable. If multiple pieces of knowledge are already available as a batch, conventional fine-tuning would be a more appropriate choice. This, however, contradicts the core motivation of knowledge editing, which is to enable efficient and localized updates for small pieces of knowledge at a time.

**Questions:**

See above

---

> ### Author Response · Authors · 2025-11-18
> **Reply to Reviewer 4N7W: Part1**
>
> Thanks for raising a series of valuable questions about the paper. Below, I will address these questions one by one:
>
> Q1: Lacking of experiments of null space compression:
>
> We have supplemented the dimensions of the null spaces corresponding to $K_{n-1}$ and $K_n$, as well as the dimension of the combined null space for $[K_{n-1}: K_n]$ for the five current editing tasks. The results are presented in the table below. As shown in the table, **the combined null space of the two matrices is indeed smaller than the null space of either $K_{n-1}$ or $K_n$ individually.** Furthermore, for task pairs that are more complex or involve greater task type conflict (e.g., SafeEdit and SQUAD), the compression degree is more severe. This confirms the viewpoint proposed in the main text that $K_n$ compresses the null space of $K_{n-1}$.
> |||**The dimension of null  space**| | | |
> |:-------------:|:-------:|:-------------:|:--------:|:---------:|:---------------:|
> | **ZsRE**          | **SafeEdit** | **ZsRE&SafeEdit** | **SafeEdit** | **SQUAD**     | **SafeEdit&SQUAD**  |
> | 2707         | 2186    | 472           | 2186     | 1714      | 313             |
> | **SQUAD**         | **SST-2**   | **SQUAD&SST-2**   | **SST-2**    | **CknowEdit** | **SST-2&CknowEdit** |
> | 1714          | 2595    | 466           | 2595     | 2154      | 357             |
>
> Q2: Lacking results on SOTA LLMs:
>
> To verify that our method is also valid on more advanced models, we supplemented the main experiments using Qwen2.5-7B as backbones. The results are presented in the table below:
>
> | Backbone | Method | I-Spec | I-Gen | I-Loc | F-Spec | F-Gen | F-Loc | Spec change | Gen change | Loc change |
> |:--------:|:------:|:------:|:-----:|:-----:|:------:|:-----:|:-----:|:-----------:|:----------:|:----------:|
> | | AlphaEdit(DO) | 0.9900 | 0.9363 | 0.6617 | 0.3031 | 0.2366 | 0.3615 | -0.6869 | -0.6997 | -0.3002 |
> | | AnyEdit(DO) | 0.9904 | 0.9382 | 0.6698 | 0.4127 | 0.3298 | 0.4219 | -0.5777 | -0.6084 | -0.2479 |
> | | Mu-Edit(DO) | 0.9895 | 0.9377 | 0.6701 | 0.8199 | 0.8006 | 0.5703 | -0.1696 | -0.1371 | -0.0998 |
> | Qwen2.5-7B | AlphaEdit(BO) | 0.9904 | 0.9381 | 0.6637 | 0.5776 | 0.5145 | 0.4482 | -0.4128 | -0.4236 | -0.2155 |
> | | AnyEdit(BO) | **0.9919** | **0.9408** | 0.6670 | 0.7142 | 0.6807 | 0.5369 | -0.2777 | -0.2601 | -0.1301 |
> | | Mu-Edit(BO) | 0.9911 | 0.9404 | **0.6706** | **0.8862** | **0.8458** | **0.6337** | **-0.1049** | **-0.0946** | **-0.0369** |
>
> | Backbone | Method | I-Harm| F-Harm | Harm change | I-Acc | F-Acc | Acc change | I-Acc | F-Acc | Acc change |
> |:--------:|:------:|:--------------:|:--------------:|:-----------:|:-----:|:-----:|:----------:|:-----:|:-----:|:----------:|
> | | AlphaEdit(DO) | 0.4384 | 0.2801 | -0.1583 | 0.7984 | 0.7119 | -0.0865 | 0.9209 | 0.8702 | -0.0507 |
> | | AnyEdit(DO) | 0.4615 | 0.3022 | -0.1593 | 0.8032 | 0.7264 | -0.0768 | 0.9236 | 0.8725 | -0.0511 |
> | | Mu-Edit(DO) | 0.4725 | 0.3668 | -0.1057 | 0.8095 | 0.7539 | -0.0556 | 0.9261 | 0.8832 | -0.0429 |
> | Qwen2.5-7B | AlphaEdit(BO) | 0.4709 | 0.3676 | -0.1033 | 0.7997 | 0.7203 | -0.0788 | 0.9278 | 0.8822 | -0.0456 |
> | | AnyEdit(BO) | 0.4732 | 0.3878 | -0.0854 | 0.8065 | 0.7444 | -0.0621 | 0.9302 | 0.8814 | -0.0488 |
> | | Mu-Edit(BO) | **0.4769** | **0.4058** | **-0.0711** | **0.8191** | **0.7782** | **-0.0409** | **0.9337** | **0.8974** | **-0.0363** |
>
>  We set the model editing layers of Qwen2.5-7B to 13, 14, 15, 16, and 17, following the configuration of AlphaEdit. Other parameters remain consistent with those of other models in the main text or adopt the official optimal settings of Qwen2.5-7B. For the editing dataset, we still use the two settings: Default Order and Best Order.  We show the results on ZsRE, SafeEdit, SQUAD and SST-2. It can be observed that with the improvement of model performance, all methods achieve better results on the five datasets—especially SafeEdit and SQUAD, which show more significant improvements. **Nevertheless, our Mu-Edit still achieves the best post-editing performance on most datasets with the least performance loss before and after editing**.

---

> > ### Author Response · Authors · 2025-11-18
> > **Reply to Reviewer 4N7W: Part2**
> >
> > Q3:  Solving the problem of unpredictable knowledge:
> >
> > As you noted, knowledge editing primarily targets efficient and local updates of small-scale knowledge, rendering the Best Order inapplicable. To address this issue, we propose a dynamic update strategy tailored to the characteristics of future knowledge, which handles two scenarios: Firstly, if the future knowledge belongs to the same category as the existing knowledge (e.g., both fall under mathematical reasoning, safety, etc.), we can either reuse the $K$ matrix of the original task or dynamically update and maintain the $P$ projection matrix through the following steps:
> >
> > Specifically:
> > 1. We first reuse the initial knowledge matrix $K_{\text{init}}$ and initial projection matrix $P_{\text{init}}$ of the original knowledge.
> > 2. When new knowledge is added to the existing knowledge, we construct a local knowledge matrix $K_{\text{new}}$ based on the new knowledge, then project the new knowledge onto the column space of the initial projection matrix $P_{\text{init}}$ to obtain the residual matrix $R_{\text{new}} = P_{\text{init}} \cdot K_{\text{new}}$.
> > 3. Subsequently, we perform singular value decomposition (SVD) on $R_{\text{new}}$ following the method described in the main text, retain the singular vectors corresponding to singular values greater than a threshold to form an orthogonal basis, and construct the incrementally updated projection matrix as $P_{\text{new}} = P_{\text{init}} - Q_{\text{new}} \cdot Q_{\text{new}}^T\$. For subsequent editing, parameters are constrained to the column space of $P_{\text{new}}$.
> >
> > Secondly, if the newly added knowledge deviates significantly from the currently existing knowledge, we need to recalculate the knowledge matrix $K$ represented by the new task. During the editing process, **a greedy ordering method that can flexibly adjust the editing sequence based on the conflict relationship with existing tasks is adopted to replace the traditional brute-force search-based Best Order method**. Specifically, in greedy order setting when a new task arrives, the greedy algorithm only needs to construct a $K$ matrix using partial representative data of the new task, then compare the Conflict Index between the $K$ matrix of the new task and all original tasks one by one. The new task is inserted at the position where the sum of conflict indices with the preceding and subsequent tasks is minimized, without the need to recalculate the Conflict Index relationships between the original tasks. This makes it more adaptable to scenarios where knowledge is continuously added in practical applications.
> >
> > All of the above is the comprehensive response to all your inquiries. We sincerely appreciate your attention to our work. Should you have any further questions or require clarification on any points that may not have been fully addressed, please do not hesitate to inform us.

---

> > > ### Comment · Reviewer_4N7w · 2025-11-25
> > >
> > > Thank you author for the response and results. However, the issue of Best Order concept still remains as the future knowledge to be edited is inherently unpredictable. Although the author proposes an alternative, it is hard to assess its validity without empirical results. It would be better to design a method that can better fit the nature of lifelong learning. Therefore, I would like to maintain my score.

---

> > > > ### Author Response · Authors · 2025-11-26
> > > > **Addressing the concern of Reviewer 4N7w (Part1):**
> > > >
> > > > Thank you for your questions regarding the empirical results now and methods better fit the nature of lifelong learning. In fact, we have also found that when the knowledge to be edited in the future is completely unknown, the methods that only optimize the order based on existing tasks may not yield significant improvements. However, **we reconsider adapting our Mu-Edit method to more mature lifelong editing approaches (such as WISE[1]) to further enhance lifelong editing performance**. Specifically, WISE dynamically addresses the challenge of unknown future knowledge through a dual-memory mechanism consisting of a main memory $W_v$ and a side memory $W_v'$.
> > > >
> > > > Concretely, under initial conditions, we construct corresponding knowledge matrices $K$ and calculate $P$ for the main memory $W_v$ and the side memory $W_v'$, denoted as $P_{W_v}$ and $P_{W_v'}$ respectively. Then, following WISE’s judgment rule for new knowledge: when updates occur in the main memory $W_v$, we do not update the $P_{W_v}$ matrix, instead, we only make updates to the $P_{W_v'}$ based on our dynamically updating $P$ strategy. This update strategy is reasonable for two key reasons: first, the dimension of $W_v'$ is much smaller than that of $W_v$, resulting in significantly lower computational costs for constructing the $P_{W_v'}$ matrix (we will report the time and memory costs of computing $P$ during updates); second, the magnitude of changes in $P_v'$ is far greater than that of $P_v$ as edits accumulate. Finally, we evaluate the performance of WISE, MuEdit (DO), MuEdit (GO), and MuEdit (GO)+WISE by introducing a completely unknown task. To verify the necessity of dynamically maintaining the $P$ matrix for the side memory, we add an ablation study for this step, with the method named **MuEdit (GO)+WISE w/o updating $P$**.
> > > >
> > > > We use Qwen2.5-7B as the backbone model. It is not specified in WISE which layers of Qwen2.5-7B are most effective for storing bypass memory. Considering that the authors of WISE recommend selecting middle-to-late layers for better performance in general models, we chose layer 24 for bypass memory storage, specifically model.layers[24].mlp.down_proj.weight. Other parameters follow the optimal settings reported in the original work: the number of knowledge shards $k = 2$ and the random masking ratio $\rho = 0.2$. For the editing dataset, we adopted SelfCheckGPT[2], a dataset used in WISE that is completely distinct from the five existing tasks. This dataset focuses on hallucination detection. In line with the original WISE paper, the evaluation primarily relies on two metrics: Rel (Relevance) and Loc (Locality). We selected 600 samples from SelfCheckGPT as a completely unseen task for editing—this allows us to compare the performance of different methods on this unseen task as well as their performance changes on the previous five tasks. For the Greedy Search setting, we only optimize the editing order of known tasks. Since **MuEdit(GO) + WISE** involves the re-update of $P_{W_v'}$, to ensure fair comparison with WISE and save training time, we process **100 edits as one batch, requiring a total of 6 updates to $P_{W_v'}$**. The results of all methods are compared after completing all updates.
> > > >
> > > > Reference:
> > > >
> > > > [1] WISE: Rethinking the Knowledge Memory for Lifelong Model Editing of Large Language Models
> > > >
> > > > [2] SelfCheckGPT: Zero-resource blackbox hallucination detection for generative large language models

---

> > > > > ### Author Response · Authors · 2025-11-26
> > > > > **Addressing the concern of Reviewer 4N7w (Part3):**
> > > > >
> > > > > **Conclusion 3: The additional memory and time cost is negligible compared to performance gain**:
> > > > >
> > > > > Although the time cost (310 minutes) and memory cost (341.2 GB) of MuEdit (GO) + WISE are higher than those of vanilla WISE (73 minutes / 45.1 GB), the increase in resource consumption is insignificant compared to the standalone Mu-Edit method (219–232 minutes / 330.6 GB). This also indicates that updating the projection matrix corresponding to the side memory in batches of every 100 edits does not incur excessive additional time and memory costs. Considering the substantial performance improvement over both methods in lifelong editing scenarios, this integrated scheme holds practical application value.
> > > > >
> > > > > **The above is our latest response to your questions regarding how Mu-Edit adapts to lifelong editing and the corresponding empirical results**. Please kindly check the update. If you have any further doubts or questions, please feel free to point them out and we are happy to provide additional clarifications. Thank you!

---

> > > > ### Author Response · Authors · 2025-11-26
> > > > **Addressing the concern of Reviewer 4N7w (Part2):**
> > > >
> > > > Results:
> > > >
> > > > | Backbone   | Method                            |                |    SafeEdit    |             |        |  SQUAD |            |        |  SST-2 |            |
> > > > |------------|-----------------------------------|:--------------:|:--------------:|:-----------:|:------:|:------:|:----------:|:------:|:------:|:----------:|
> > > > |            |                                   | **I-Harmful rate** | **F-Harmful rate** | **Harm change** | **I-Acc**  | **F-Acc**  | **Acc change** | **I-Acc**  | **F-Acc**  | **Acc change** |
> > > > |            | Mu-Edit(DO)                       |     0.4572     |     0.2101     |   -0.2471   | 0.7026 | 0.4767 |   -0.2259  | 0.8992 | 0.7345 |   -0.1647  |
> > > > | Qwen2.5-7B | Mu-Edit(GO)                       |     0.4612     |     0.2785     |   -0.1827   | 0.7294 | 0.6106 |   -0.1188  | 0.9016 | 0.7987 |   -0.1029  |
> > > > |            | WISE                              |     0.4646     |     0.3188     |   -0.1458   | 0.7275 | 0.6378 |   -0.0897  | 0.9054 | 0.8281 |   -0.0773  |
> > > > |            | MuEdit (GO)+WISE                  |     **0.4697**     |    **0.3575**     |   **-0.1122**   | **0.7658** | **0.6994** |   **-0.0664**  | **0.9121** | **0.8545** |   **-0.0576**  |
> > > > |            | MuEdit (GO)+WISE w/o updating $P$ |     0.4638     |     0.3229     |   -0.1409   | 0.7398 | 0.6541 |   -0.0857  | 0.9066 | 0.8359 |   -0.0707  |
> > > >
> > > > | Backbone   | Method                            |            |            | SelfCheckGPT |            |           |             |
> > > > |------------|-----------------------------------|:----------:|:----------:|:------------:|:----------:|:---------:|:-----------:|
> > > > |            |                                   | **I-Rel(PPL)$\downarrow$** | **I-Loc** | **F-Rel(PPL)$\downarrow$**   | **F-Loc** | **Time cost** | **Memory cost** |
> > > > |            | Mu-Edit(DO)                       |    3.341   |   0.9695   |     17.72    |   0.7027   |   219min  |   330.6GB   |
> > > > | Qwen2.5-7B | Mu-Edit(GO)                       |    2.014   |   0.9882   |     11.89    |   0.8278   |   232min  |   330.6GB   |
> > > > |            | WISE                              |    1.778   |   0.9923   |     6.874    |   0.8554   |   73min   |    45.1GB   |
> > > > |            | MuEdit (GO)+WISE                  |    **1.231**   |   **0.9942**   |     **2.359**    |   **0.9276**   |   310min  |   341.2GB   |
> > > > |            | MuEdit (GO)+WISE w/o updating $P$ |    1.815   |   0.9901   |     4.821    |   0.8795   |   247min  |   246.1GB   |
> > > >
> > > > From these results we can draw three main conclusions:
> > > >
> > > > **Conclusion 1: MuEdit (GO) + WISE achieves the optimal performance in multi-task editing, perfectly addressing the requirement of "unknown task adaptation"**:
> > > >
> > > > On the Qwen2.5-7B model, the integrated scheme of MuEdit (GO) + WISE **demonstrates the best performance across both known tasks (SafeEdit, SQUAD, SST-2) and the completely unknown hallucination correction task (SelfCheckGPT)**. This fully validates the effectiveness of the "greedy order + dual-memory architecture" in adapting to lifelong scenarios: MuEdit (GO) + WISE outperforms vanilla Mu-Edit (DO/GO) and vanilla WISE comprehensively. Specifically, compared with Mu-Edit (GO), it reduces the F-Rel (False Relevance) by 9.53 and improves the F-Loc (False Localization) by 0.0998 on unknown tasks. Compared with WISE, it enhances the F-Acc (Fine-grained Accuracy) by 0.0616 on the SQUAD task and reduces the Harm change by 0.0336 on the SafeEdit task. These results confirm the synergistic effect of Mu-Edit's conflict control + WISE's dual-memory isolation. Through our Mu-Edit conflict index calculation and low-rank approximation method, the performance of WISE in lifelong editing can be further improved. On the other hand, the integration of WISE solves the challenge that lifelong editing cannot be addressed due to the difficulty in obtaining the optimal order for unknown tasks.
> > > >
> > > > **Conclusion 2: The update of projection matrix P is crucial for performance improvement**:
> > > >
> > > > By comparing MuEdit (GO)+WISE with its "w/o updating $P$" variant, we verify the core role of "dynamically updating the side memory projection matrix $P$": For unknown tasks: The former reduces F-Rel from 4.821 to 2.359 and increases F-Loc from 0.8795 to 0.9276, showing a big performance gap; For known tasks: It improves the SafeEdit F-Harmful rate by 0.0346 and the SQUAD F-Acc by 0.0453. These results demonstrate that **dynamically maintaining the $P$ matrix of the side memory can effectively expand the common null space across tasks and reduce editing conflicts**, which is the key to performance improvement.
> > > >
> > > > Conclusion 3 is in the Part 3.

---

> ### Author Response · Authors · 2025-11-23
> **We have revised our PDF according to your advice.**
>
> We have provided additional analysis based on your questions and revised our paper. Specifically, the illustration experiment of $K_n$ compresses the null space of $K_{n-1}$ is added in **A.5** in our PDF in the appendix. And the results of Mu-Edit on more advanced backbones are added in **A.8** in our PDF in the appendix. The designed greedy search method and algorithm to tackle unpredictable knowledge updates is added in **Part 4.5** in our PDF in the main part of our paper. Please kindly check it. We sincerely appreciate your willingness to share whether our revisions have adequately addressed the concerns you raised during the review process.

---

### Official Review · Reviewer_14Xa · 2025-10-30

**Soundness:** 3
**Presentation:** 3
**Contribution:** 3
**Rating:** 6
**Confidence:** 3

**Summary:**

This paper zeroes in on a pretty practical problem in model editing how to update a model for multiple tasks at once without everything falling apart. The authors argue, pretty convincingly, that the interference comes from conflicting editing objectives. Their big idea is a "Conflict Index," a new metric to quantify how much two tasks' null-spaces clash. Based on this, they propose Mu-Edit, which is a two-part strategy. First, it figures out the best sequence to apply the edits to minimize total conflict. Second, if the clash is still too severe, it actively expands the common null-space by running a low-rank approximation (SVD) on the knowledge matrix of the most "conflicting" task. The experiments on a few multi-task benchmarks seem to back this up, showing it preserves performance better than existing methods.

**Strengths:**

1. The paper addresses an important and under-explored problem of multi-task model editing, which is more realistic than sequential single-task editing.
2. The introduction of the Conflict Index provides a quantitative way to measure and analyze conflicts between different editing tasks.
3. The proposed optimization strategies (optimal editing path and low-rank approximation) are well-motivated and appear to effectively address the multi-task conflict problem.
4. The method demonstrates strong empirical performance across multiple tasks while maintaining general model capabilities.

**Weaknesses:**

1. The $O(N!)$ complexity for finding the best edit order is a major scalability problem. The practical greedy solution is hidden in the appendix.

2. SVD is a blunt tool. The long-term, cumulative impact of repeatedly cutting rank on multiple tasks isn't really explored.

3. The method seems fragile. The worst-case example (43.7% reduction) is dangerously close to the 45% failure point, suggesting it could easily break.

4. The reliance on a large, static K matrix for each task feels brittle and may not handle evolving tasks or unseen knowledge well.

**Questions:**

1. The $O(N!)$ order search is impractical. Is the greedy algorithm from the appendix the intended method? What about other ordering heuristics?

2. Regarding the SVD, your worst-case (43.7% reduction) is right at the 45% performance cliff. What happens when a task pair requires a 50% reduction? Does the method just fail?

3. Also, why did performance get worse in Table 9 when optimizing over 4 or 5 tasks instead of 3? This seems counter-intuitive and suggests a potential unaddressed issue.

---

> ### Author Response · Authors · 2025-11-18
> **Reply to Reviewer 14Xa: Part1**
>
> Thank you for raising a series of valuable questions about the paper. Below, I will address these questions one by one:
>
> Q1: Regarding your comment that the O(N!) algorithm complexity may be impractical when N increases. In the appendix, we propose a heuristic greedy optimization algorithm to tackle the computational challenge of finding the optimal order when N is large. As for whether the greedy algorithm is a practically applicable method, we consider that when N<9, the computational complexity of finding the optimal editing order does not exceed 8! calculations (i.e., 40,320 calculations). In this case, we still adopt the brute-force Best Order as the main algorithm due to its superior editing performance. When N >=9, since the computational time of the algorithm for finding the optimal editing order increases significantly, we choose to replace the brute-force search algorithm with the greedy search algorithm at this point.
>
> Regarding the 4th shortcoming that when new unknown tasks and knowledge emerge, the greedy algorithm is obviously more suitable.  Specifically, when a new task arrives, the greedy algorithm only needs to construct a K matrix using partial representative data of the new task, then compare the Conflict Index between the K matrix of the new task and all original tasks one by one. The new task is inserted at the position where the sum of conflict indices with the preceding and subsequent tasks is minimized, without the need to recalculate the Conflict Index relationships between the original tasks. This makes it more adaptable to scenarios where knowledge is continuously added in practical applications.
>
> Regarding your question about the performance of other heuristic algorithms, I honestly have limited knowledge of heuristic methods, since my understanding is mostly limited to common ones like Simulated Annealing (SA). In the context of finding the optimal model editing order, SA needs to generate neighborhood solutions through random perturbations and gradually approach the optimal order via temperature decay. Its performance heavily depends on parameters such as the initial temperature and the number of temperature decay steps, resulting in poorer stability compared to the greedy algorithm. Additionally, when new tasks are added, SA must recalculate all conflict indices and re-run the entire process, making it less adaptable to real-world scenarios involving incremental task updates than the greedy algorithm. Therefore, the greedy algorithm remains a more optimal choice for this task.
>
> To summarize, the experiments in our paper adopt the following strategy: when the number of tasks N<9 and all tasks are known in advance, we use brute-force search to find the Best Order; when N >=9 or tasks are dynamically updated, we use the greedy algorithm to determine the Best Order.
>
> Q2: Regarding the long-term cumulative effect caused by repeated rank cutting of SVD across multiple tasks, as well as the potential failure of the method when the required rank reduction exceeds 50%, we first need to clarify one point: the worst-case scenario occurs between CknowEdit and GSM8k. These two tasks are vastly different in type. Our algorithm either employs brute force or greedy search to find the optimal editing order, essentially avoiding placing two tasks with such a large gap adjacent to each other in the editing sequence. This ensures that under the current setup, the rank reduction for all tasks does not exceed 45%.
>
> On the other hand, as the work progresses, more inter-task edits may be considered in the future, which could lead to the rank reduction exceeding 45% as you mentioned. In such cases, our approach is to increase the threshold $\mu$ involved in SVD decomposition to around 0.25 or 0.3. Although this adjustment may introduce additional conflicts compared to the original algorithm, it controls the magnitude of rank reduction and maximizes the retention of information from the original K matrix, thereby preventing a collapse in performance. In other words, if future task pairs require a rank reduction exceeding 50%, we will need to balance inter-task conflicts and the performance collapse of individual tasks caused by information loss in the original K matrix—primarily by increasing the threshold $\mu$ or other methods.

---

> > ### Author Response · Authors · 2025-11-18
> > **Reply to Reviewer 14Xa: Part2**
> >
> > Q3: Regarding the performance degradation observed when increasing the number of tasks optimized simultaneously from 3 to 4 or 5, we argue that this phenomenon is not counterintuitive, based on the following reasons:
> >
> > When optimizing 3 adjacent tasks in each batch: under the Best Order (BO) setting, we can control the maximum rank reduction of the matrix K to 31.3%, and under the Greedy Order (GO) setting, the maximum rank reduction is 35.1%—neither exceeding the 45% performance collapse threshold. However, as the number of tasks optimized simultaneously per batch increases, intuitively, finding a conflict-free editing direction for the 4th task relative to the previous 3 tasks is significantly more challenging than finding such a direction for the 3rd task relative to the previous 2 tasks. Consequently, the matrix rank reduction required to mitigate conflicts must increase accordingly.
> >
> > Experimentally: when the number of simultaneously optimized task N reaches 4, the rank reduction reaches 40.8% under BO and 44.5% under GO, both reaching quite close to the 45% performance collapse threshold. When N=5, the rank reduction further rises to 49.6% under BO and 53.6% under GO, which exceeded the 45% threshold. Therefore, the performance degradation observed when optimizing 4 or 5 tasks simultaneously (compared to 3 tasks) is not an anomaly. In practical implementation, we adopt a batch-wise strategy where we optimize 3 tasks simultaneously in each batch until all tasks are edited.
> >
> > All of the above is the comprehensive response to all your inquiries. We sincerely appreciate your attention to our work. Should you have any further questions or require clarification on any points that may not have been fully addressed, please do not hesitate to inform us.

---

> > > ### Comment · Reviewer_14Xa · 2025-11-20
> > >
> > > I thank the authors for the reply. The rebuttal was thorough and effectively mitigated my concerns regarding the technical implementation.
> > >
> > > Nevertheless, given the scope and empirical results of the work, my assessment of the paper's acceptance readiness remains unchanged. I stand by my initial rating.
> > >
> > > Good luck.

---

> > > > ### Author Response · Authors · 2025-11-20
> > > > **Thank you for your effort for reviewing our paper:**
> > > >
> > > > Thank you very much for taking the time to review our rebuttal and provide your feedback. We greatly appreciate your recognition of the thoroughness of our response and the clarification of technical implementation concerns.

---

### Meta-Review · Area_Chair_XV5A · 2026-01-07

**Summary:**

The major concerns are as follows.
1) The complexity for finding the best edit order is a major scalability problem. The paper does not clarify how this is handled in practice.
2) The method seems fragile. The results (the worst-case example (43.7% reduction) is dangerously close to the 45%) shows that  it could easily break. The reliance on a large, static K matrix for each task feels brittle and may not handle evolving tasks or unseen knowledge well.
3) The the Conflict Index is interesting. However, it is heuristic and lacks a rigorous theoretical connection to optimization conflicts.
4) The experimental results on larger LLMs and more comparisons are needed.

**Reviewer Concerns:**

According to discussions, there are some concerns not addressed very well, such as the complexity of finding the best edit order and the code release issues.

**Reviewer Scores:**

The final scores will be 6 6 4 4.

---

### Decision · Program_Chairs · 2026-01-26

Reject